# IMPROVING REAL-WORLD SEQUENCE DESIGN WITH A SIMPLE META-HEURISTIC FOR DETECTING DISTRIBUTION SHIFT

## ABSTRACT

Biological sequence design is one of the most impactful areas where model-based optimization is applied. A common scenario involves using a fixed training set to train predictive models, with the goal of designing new sequences that outperform those present in the training data. This by definition results in a distribution shift, where the model is applied to samples that are substantially different from those in the training set (or otherwise they wouldn't have a chance of being much better). While most MBO methods offer some balancing heuristic to control for false positives, finding the right balance of pushing the design distribution while maintaining model accuracy requires deep knowledge of the algorithm and artful application, limiting successful adoption by practitioners. To tackle this issue, we propose a straightforward meta-algorithm for design practitioners that detects distribution shifts when using any MBO. By doing a real-world sequence design experiment, we show that (1) Real world distribution shift is far more severe than observed in simulated settings, where most MBO algorithms are benchmarked (2) Our approach successfully reduces the adverse effects of distribution shift. We believe this method can significantly improve design quality for sequence design tasks and potentially other domain applications where offline optimization faces harsh distribution shifts.

## 1 INTRODUCTION

Machine learning (ML) is increasingly guiding design in fields like materials science, chemistry, and biology (Gómez-Bombarelli et al., 2018; Alley et al., 2019; Wu et al., 2019; Wang et al., 2023). ML in design is used to find inputs that enhance specific properties, such as designing peptides with better antimicrobial properties (Gupta & Zou, 2019), or optimizing superconductors for higher critical temperatures (Fannjiang & Listgarten, 2020). Model-free design methods often rely on many rounds of expensive, time-consuming experiments (Arnold, 1998). ML-guided design reduces these costs by experimental steps with model predictions (Yang et al., 2019).

Offline model-based optimization (MBO) (Trabucco et al., 2022; Angermueller et al., 2019; Linder et al., 2020) a key paradigm in ML-guided sequence design (Sinai & Kelsic, 2020). It involves creating a "surrogate" model from a set of input-property pairs to predict property values. Then, one searches the space of possible input values to find a new batch of inputs with predicted property values that are greater than those observed in the training set. This search can use various optimization methods including evolutionary algorithms for discrete inputs or gradient-based methods for continuous inputs (Sinai et al., 2020; Trabucco et al., 2022; Angermueller et al., 2019).

MBO is conceptually simple and versatile, applicable to various design problems with black-box methods, given an initial dataset. However, its effectiveness depends on careful implementation due to several challenges, notably the tendency of black-box search methods to focus on out-of-distribution (OOD) regions compared to the training data (Fannjiang & Listgarten, 2023). This "distribution shift" can lead to unreliable predictions when provided with OOD inputs, especially for high-capacity neural network models. In extreme cases, the search method might exploit these unreliable predictions, especially in maximization problems, resulting in nonsensical outputs that do not translate to real world results (Brookes et al., 2019).

The main challenge in effective MBO is detecting distribution shifts from training data and identifying resulting false-positive (adversarial) examples. Several methods have been proposed to minimize these shifts, including using Bayesian Optimization (BO) to regularize the search based on the surrogate model's predictive uncertainty (Snoek et al., 2012), limiting the search to regions with high likelihood in the training distribution (Brookes et al., 2019; Linder et al., 2020; Angermueller et al., 2019), and incorporating domain-specific knowledge into the search algorithm (Sinai et al., 2020). While these methods can reduce the problem, they do not fully eliminate distribution shift and can be challenging to adapt to specific problems. Critically, most such methods are only tested in simulated settings.

A different emerging direction in reducing distribution shift is the use of foundation models trained on all known proteins (Madani et al., 2023; Lin et al., 2023), where the set of data in the training domain is expansive. However, while they are the go-to choice for zero-shot generation, design with these models is nascent, and they are not better than supervised approaches when some functional data has already been collected (Dallago et al., 2021).

Finally, the complexity of using MBO algorithms correctly (e.g. when using RL or generative models), limits the adoption among practitioners. For instance, selecting a trust-region for any search algorithm can be an art rather than science, and risks wasting experimental resources.

In this work, propose a simple meta-heuristic for detecting distribution shifts in MBO and correcting it, reducing the need to fine tune algorithms precisely within the MBO loop. Our approach can be combined with any existing oracle-based design method to further minimize adversarial designs. Specifically, we propose training a binary classifier to distinguish between the initial training samples and a separate set of samples drawn from one's chosen search algorithm. We demonstrate that the logit score output by the trained "OOD classifier" is an interpretable and effective metric for determining which designed sequences are OOD and thus are associated with unreliable predictions from the surrogate model. This method is straightforward to implement given the initial training set and a search method, and can be used in conjunction with any black-box surrogate model and search method. We suggest multiple ways in which the OOD classifier score (henceforth referred to as "OOD scores") can be used to guide the selection of designed inputs for subsequent experimental verification.

We study the OOD classifier in three increasingly realistic problems. The first is an illustrative toy problem with a two-dimensional input space. Next, we validate the method in a simulated environment for protein structure prediction of a small protein, where we can use *in silico* structure prediction methods to measure the effectiveness of our black-box design. Finally, we apply our method in a real-world experiment where we design Adeno-Associated Virus (AAV) capsid seqequences, a complex protein of major importance to gene therapy. In this case, we use AdaLead as our MBO algorithm because its best published MBO algorithm for AAV design (Bryant et al., 2021; Sinai et al., 2020). We trace the generated sequences over the course of MBO trajectories where they are experimentally tested for multiple important properties. This unique dataset allows us to track the extent and effects of distribution shift during MBO, and test how methods such as the OOD classifier are able to detect such a shift. We show that the OOD classifier can improve the outcome in this black-box design problem, which is inaccessible to structural and domain-informed approaches. Importantly, we observe that the simulated settings (including our second objective and additional ones we test in the supplement) show much weaker distributions shift.

## 2 METHODS

### 2.1 OFFLINE MODEL BASED OPTIMIZATION

The objective in a data-driven design problem is to identify an optimal input $\boldsymbol{x}^*$ of some ground-truth function, $f(\boldsymbol{x})$, that encodes a scalar property of the input. This can be expressed as the objective $\boldsymbol{x}^* = \arg\max_{\boldsymbol{x} \in \mathcal{X}} f(\boldsymbol{x})$, where $\mathcal{X}$ is a bounded set that we refer to as the "input space" of the problem. The input space may be a discrete or continuous space, and $f(\boldsymbol{x})$ is typically assumed to be a black box from which we can only make zeroth-order evaluations. We assume the availability of a static dataset, $D = \{(\boldsymbol{x}_i, y_i)\}_{i=1}^N$, consisting of elements of the input space paired with corresponding noise-corrupted evaluations of the ground truth function.

In MBO, the design strategy is to train a surrogate model $\hat{f}(\boldsymbol{x})$ on the dataset $D$ using an appropriate supervised regression strategy (e.g. minimizing the mean squared error of predictions from the model via stochastic gradient descent). This surrogate model is then used to guide a search around the input space to find inputs with high predicted property values. This search may take the form of an optimization algorithm (Sinai et al., 2020; Angermueller et al., 2019), probabilistic sampling method (Brookes et al., 2019), or sampling from a generative model (Gómez-Bombarelli et al., 2018; Linder et al., 2020; Nijkamp et al., 2022).

In an ideal scenario for MBO, the inputs of the training dataset would be evenly distributed across the input space, and therefore the error between the surrogate model and the ground truth function would be roughly equal in all regions of input space. In such a case, one could safely optimize the surrogate model directly with a reasonable assumption that this would produce an input that is close to a local optimum of the ground truth function. In most practical scenarios, however, the training inputs are not evenly distributed, and are instead concentrated in a small region of input space. In such cases, the error of the surrogate model will change drastically depending on which region of input space is being tested. Further, search methods that use the surrogate model will tend to move to regions of input space where there is a low density of training points, and therefore predictions from the surrogate model may be unreliable (Brookes et al., 2019; Fannjiang & Listgarten, 2023). In other words, the design strategy induces a distribution shift between the training distribution and the "design distribution" that results from performing the search (Fannjiang et al., 2022; Wheelock et al., 2022).

In the next section, we discuss the type of distribution shift commonly observed in design problems and how we may be able to detect which inputs are most affected by this shift.

## 2.2 Distribution shift in design

Distribution shift in sequence design problems typically takes the form of *covariate shift*, in which $p_{\text{tr}}(\boldsymbol{x}) \neq p_{\text{te}}(\boldsymbol{x})$, where $p_{\text{tr}}(\boldsymbol{x})$ and $p_{\text{te}}(\boldsymbol{x})$ are the distributions of training and test inputs, respectively (Shimodaira, 2000). Under covariate shift, the ground truth conditional distribution of property values given inputs, $p(y|\boldsymbol{x})$, remains fixed between the train and test distributions. This is because design problems are oftentimes modeling an underlying system that has a fixed relationship between $\boldsymbol{x}$ and y, e.g., a biological process.

In MBO, the search method uses information from the surrogate model to guide its exploration of input space. This induces a dependence between the training distribution and the distribution of designed inputs, $p_{\text{de}}(\boldsymbol{x})$, resulting in a form of covariate shift known as "feedback covariate shift" that can have a particularly pernicious effect on the accuracy of model predictions (Fannjiang et al., 2022; Stanton et al., 2023). Although our proposed method can be used to detect any type of covariate shift, our experiments are focused on demonstrating that it is effective in the difficult case of feedback covariate shift.

Figure 1 illustrates feedback covariate shift in MBO, detailed in Section 4.3. Here, a discrete optimization method optimized a surrogate model to identify protein sequences with high predicted values for a desired property. We tested sequences from each point along the optimization trajectory in a bulk physical experiment to (i) determine whether they satisfy basic functional properties, and thus not adversarial, and (ii) measure the actual property value. Figure 1a shows increasing surrogate model predictions along the optimization trajectory, as expected. However, Figure 1b demonstrates a significant increase in prediction error along the trajectory compared to true property values, indicating a distribution shift during design. This increase in error occurs *despite the model's low MSE on a random holdout* as shown in the inset of Figure 1b. Figure 1c further shows that later designs in the optimization trajectory mostly consist of adversarial sequences, which fail to meet basic requirements of a functional protein, highlighting the severe distribution shift.

Our central aim is to detect when a covariate shift such as that shown in Figure 1 has occurred, so that input points associated with reliable predictions from the surrogate model can be identified. This allows one to avoid selecting inputs for experimental validation that may have unreliable predictions. In order to detect such shifts, we require a score, $s(\boldsymbol{x})$ that reports the extent, or "intensity" of the shift at each input point. An intuitive score that has been frequently used is the density ratio between the test distribution and train distribution (Sugiyama et al., 2007). In the design setting a fixed test is not given, so we select the unlabeled design distribution as our test distribution (see Appendix D for

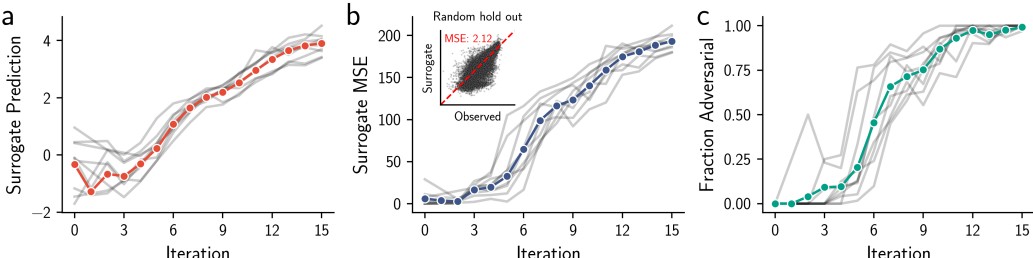

Figure 1: Distribution shift in a protein design problem. Independent trajectories of discrete iterative optimization were run using a surrogate model of a desired property as the objective function and sequences generated along the trajectory were evaluated experimentally. Each subplot shows statistics associated with these trajectories; gray lines correspond to individual trajectories and colored lines display the mean values over all trajectories. (a) Predicted property value from the surrogate model. (b) MSE between surrogate predictions and observed experimental measurement of the property. Inset shows surrogate predictions versus observed property values for a randomly held out set of data from the training distribution. (c) Fraction of sequences observed to be adversarial examples.

comparisons to other design-independent test sets).

$$s(\boldsymbol{x}) = \frac{p_{\text{de}}(\boldsymbol{x})}{p_{\text{tr}}(\boldsymbol{x})}. \tag{1}$$

For the purpose of detecting covariate shifts that induce error in the surrogate model, Equation 1 is not necessarily the ideal score. In particular, one should expect $1/p_{\text{tr}}(\boldsymbol{x})$ to be correlated with the error of the surrogate model at a point $\boldsymbol{x}$, but the density of the design distribution should not necessarily impact the surrogate error. However, estimating the density of a high-dimensional distribution such as $p_{\text{tr}}(\boldsymbol{x})$ can be difficult in practice (Hido et al., 2011; Weinstein et al., 2022), while we will see that the density ratio $s(\boldsymbol{x})$ is simple to estimate using a binary classifier. Further we provide a simple argument in Appendix B to suggest that the $1/p_{\text{tr}}(\boldsymbol{x})$ term will tend to dominate the difference in scores between two nearby designed inputs; therefore the distinction between $s(\boldsymbol{x})$ and $1/p_{\text{tr}}(\boldsymbol{x})$ is small in practice and both can be used to detect distribution shift that results in surrogate model error.

## 2.3 DISTRIBUTION SHIFT DETECTION WITH BINARY CLASSIFICATION

It is well known that a binary classifier can be trained such that its output values approximate a density ratio such as that in Equation 1. In particular, let $g(\boldsymbol{x})$ be a model with a single continuous scalar output, and let $\rho(\boldsymbol{x}) = \sigma(g(\boldsymbol{x}))$ be the probability score of such a model, where $\sigma$ represents the sigmoid function. Let $p_{\text{de}}(\boldsymbol{x})$ and $p_{\text{tr}}(\boldsymbol{x})$ represent the distribution of positive and negative class inputs, respectively. Then, the binary cross-entropy loss to distinguish between these two classes is:

$$L(\rho) = \mathbb{E}_{p_{\text{tr}}(\boldsymbol{x})}[-\log(1 - \rho(\boldsymbol{x}))] + \mathbb{E}_{p_{\text{de}}(\boldsymbol{x})}[-\log \rho(\boldsymbol{x})] \tag{2}$$

We follow Srivastava et al. (2023) and take the functional derivative of $L$ with respect to $\rho$, resulting in

$$\frac{\delta L}{\delta \rho} = \frac{p_{\text{tr}}(\boldsymbol{x})}{1 - \rho(\boldsymbol{x})} - \frac{p_{\text{de}}(\boldsymbol{x})}{\rho(\boldsymbol{x})} \tag{3}$$

Setting this expression equal to zero demonstrates that the minimizer of the binary cross entropy can be used to exactly predict the OOD score $s(\boldsymbol{x})$:

$$s(\boldsymbol{x}) = \frac{p_{\text{de}}(\boldsymbol{x})}{p_{\text{tr}}(\boldsymbol{x})} = \frac{\rho(\boldsymbol{x})}{1 - \rho(\boldsymbol{x})}. \tag{4}$$

The equivalence shown in Equation 4 is often referred to as the "density ratio trick" and provides a straightforward path towards estimating the OOD scores $s(\boldsymbol{x})$. In practice, we cannot exactly minimize Equation 2 with respect to $\rho$, so we parameterize $g$ with parameters $\boldsymbol{\theta}$, and minimize Equation 2 with respect to $\boldsymbol{\theta}$. We then calculate approximate OOD scores as:

$$\hat{s}(\boldsymbol{x}) = \frac{\rho_{\boldsymbol{\theta}}(\boldsymbol{x})}{1 - \rho_{\boldsymbol{\theta}}(\boldsymbol{x})} \tag{5}$$

where $\rho_{\boldsymbol{\theta}}(\boldsymbol{x}) = \sigma(g_{\boldsymbol{\theta}}(\boldsymbol{x}))$ are the probability scores associated with the parameterized model $g_{\boldsymbol{\theta}}(\boldsymbol{x})$. In all of our experiments, $g_{\boldsymbol{\theta}}$ is instantiated as either a multi-layer perceptron (MLP) or convolutional neural network (CNN); however, any model architecture could be used within this framework.

## 2.4 Selection using OOD scores

The final step of any MBO procedure is selecting one or more inputs sampled from the design distribution for follow-up evaluation on the ground truth function (i.e. testing the inputs' property values in a physical experiment). We propose using the approximate OOD scores calculated via Equation 5 to guide this selection process. Our experiments will demonstrate these OOD scores provide an interpretable metric for detecting distribution shift in MBO methods, and can reliably identify regions of input space where we can expect a surrogate model to make accurate predictions. Given a set of designed inputs, a distribution shift-aware selection procedure should prioritize inputs associated with high surrogate model predictions that are also expected to be accurate based on OOD scores. How exactly a practitioner chooses to enforce this intuition will ultimately be application-specific, and should be tailored to the user's needs and level of risk-tolerance. Three possible techniques for doing so are (i) a cutoff process, where only allows an input to be selected if $\rho_\theta(\boldsymbol{x}) < c$ for a chose cutoff value $c$, (ii) stratified selection in which selects a specified number of inputs from ranges of OOD scores (e.g. select ten inputs from scores between $a$ and $b$, and 10 inputs from scores between $b$ and $c$) and (iii) selection based on a user-defined utility function.

## 2.5 Deploying offline MBO in the wild

The primary aim of creating new offline MBO algorithms is to deploy them in real-world scenarios. However, methods development faces a gap between simulation and the real world. Typically, methods development is relegated to simulation, and in the rare cases where real-world deployment is involved, the goal is to use the method to produce an optimized design. We are aware of no instances where offline MBO has been applied in real-world situations specifically to study the algorithm itself. Studying offline MBO in real-world settings is essential for improving algorithm development and understanding where simulations break down. The experimental setup for analyzing offline MBO retrospectively can differ in crucial aspects from the usual optimization process aimed at improving a design. Consider the following example. The distribution of designed inputs will gradually shift away from the training data with each optimization step. To study this shift and to determine the limitations of offline MBO, it is crucial to label data at every step to understand where our predictions become inaccurate. On the other hand, if the sole objective is to optimize, there is no need to waste valuable resources on assessing inputs that likely don't improve design. Herein, we design a real-world dataset to study distribution shifts and to evaluate our method in the wild. In the related work section, we discuss the state of datasets in offline MBO and some of its limitations.

## 3 Related Work

While our approach is not an MBO algorithm, it is closely related and meant as a complement to such algorithms. We cover related work in this spirit.

**Regularized search in ML-guided design** The OOD classifier, inspired by ML-guided design, limits search to regions near the training distribution. "Latent space optimization" is one method, involving an encoder-decoder model jointly trained with a regression model that maps latent space points to property values. After training, optimization in the latent space identifies points with high property values for generation. This optimization remains close to the training distribution through a spherical boundary (Gómez-Bombarelli et al., 2018) or by implicitly altering the training objective(Castro et al., 2022). Another method is adaptive generative modeling, where model parameters are updated iteratively to generate inputs with high model scores. This approach stays close to the training distribution by weighting points based on their training distribution density (Brookes et al., 2019) or through gradual weight updates (Gupta & Zou, 2019). Unlike these methods, which depend on specific surrogate models or search strategies, the OOD classifier is versatile, compatible with any surrogate model or search technique.

Fannjiang et al. (2022) proposes an MBO method that applies conformal prediction to FCS, ensuring search is limited to inputs where the surrogate model is guaranteed to be reliable. Stanton et al. (2023) applies these methods to Bayesian Optimization (Snoek et al., 2012). While effective against FCS, these strategies are computationally intensive as they require training at least $n$ models, where $n$ is the number of test data points. This is impractical for high-throughput sequence design problems. In contrast, the OOD classifier method is more scalable, requiring only one model for a fixed test set. It

is accessible to practitioners with basic machine learning knowledge and our results will indicate that it can also detect FCS.

**Density ratio estimation for covariate shift and outlier detection** The density ratio between test and train distributions (e.g., Equation 1), has been widely used to address covariate shift in supervised and reinforcement learning. In supervised learning, when the test set is shifted from the training set, the density ratio, $w(\boldsymbol{x}) := p_{\text{te}}(\boldsymbol{x})/p_{\text{tr}}(\boldsymbol{x})$, can be used to minimize a loss function, $\ell(\boldsymbol{x}, y)$, averaged over a test set based on the equivalence $\mathbb{E}_{p_{\text{te}}(\boldsymbol{x}, y)}[\ell(\boldsymbol{x}, y)] = \mathbb{E}_{p_{\text{tr}}(\boldsymbol{x}, y)}[w(\boldsymbol{x})\ell(\boldsymbol{x}, y)]$ (Shimodaira, 2000; Sugiyama et al., 2007). This led to techniques for estimating $w(\boldsymbol{x})$ using binary classifiers and the density ratio trick (Sugiyama et al., 2012). Propensity scores are also closely related, which learns a weight equal to the Radon-Nikodym derivative (RND) between the test set $q$ and the training set $p$ $w(\boldsymbol{x}) := dp_{\text{te}}(\boldsymbol{x})/dp_{\text{tr}}(\boldsymbol{x})$ (Agarwal et al., 2011). Similar to the OOD classifier, binary classification models are used to estimate the propensity score. In reinforcement learning, the density ratio helps regularize policies in off-policy and offline RL (Precup et al., 2000), reweighting marginal state-action pairs during training to avoid low-support regions. However, while these techniques address non-feedback covariate shift, they can not be applied in the feedback case because this requires the surrogate model to already be trained in order to generate the test distribution, $p_{\text{de}}(\boldsymbol{x})$. Thus, we apply the density ratio score in Equation 1 to detect covariate shift of designed inputs instead of weighting samples during training.

Our use of OOD scores at test time is most similar to how density ratio estimates are used to detect outliers in a test set, such as by Hido et al. (2011). Our methods and results differ from this work in a few notable ways. First, we use deep binary classifiers in order to estimate density ratios, in contrast to the linear models used in Hido et al. (2011). Further, the outlier detection task only requires detection of standard (non-feedback) covariate shift; in contrast, our result demonstrates the density ratios can be used to mitigate the effects of feedback covariate shift, and thus can be used for ML-guided design problems. We also find that the use of a design-induced test set, rather than a fixed test set is crucial to the performance of our method (see Appendix D for a comparison of $q$ distributions). Finally, our results show that OOD scores can be used as a continuous measure that reports on the degree of distribution shift intensity at any point in input space, rather than only for the binary classification task of outlier detection. It is also notable that these approaches have never been applied in sequence design, and their effectiveness in practice has not been established before.

**Datasets in sequence design** While the ML-guided sequence design field has produced a wide variety of datasets and benchmarks in recent years, there remains a gap in understanding how offline MBO methods will perform in the real-world given their performance in these simulated settings (see Appendix F for experimental evidence of this gap). Some examples of progress in dataset development and benchmarking include the following. In protein engineering, one of the more active subdomains where offline MBO has been applied, FLIP (Dallago et al., 2021) curated published data and developed tasks and metrics for model generalization, emphasizing dataset-splitting techniques to probe generalization for offline static datasets. There has also been efforts towards producing comprehensive (i.e., combinatorially complete) low-dimensional datasets (Poelwijk et al., 2019). These datasets are useful for evaluating supervised model performance, but are not suitable for evaluating design methods. Outside of protein engineering, Design-Bench (Trabucco et al., 2022) consolidates offline MBO challenges across problem domains, allow for algorithm evaluation in a variety of simulated contexts. There exists very few examples of real-world evaluation of design strategies, and none that explicitly study distribution shift (Bryant et al., 2021; Madani et al., 2023). In the next section, we will use a two-dimensional example to illustrate our method and then evaluate its performance in a real-world deployment of offline MBO.

## 4 RESULTS

### 4.1 2D TOY MODEL

We first demonstrate the utility of the OOD classifier in a two-dimensional toy problem. The goal here is to learn a surrogate model of a ground truth function, $f(\boldsymbol{x})$, and to determine the regions in input space where the surrogate model's predictions are reliable. We employ a modified Himmelblau function Himmelblau (1972) as the ground truth. This function is negated and normalized such that all function values are between 0 and 1 in the range $[-5, 5]$ of both input dimensions. The Himmelblau

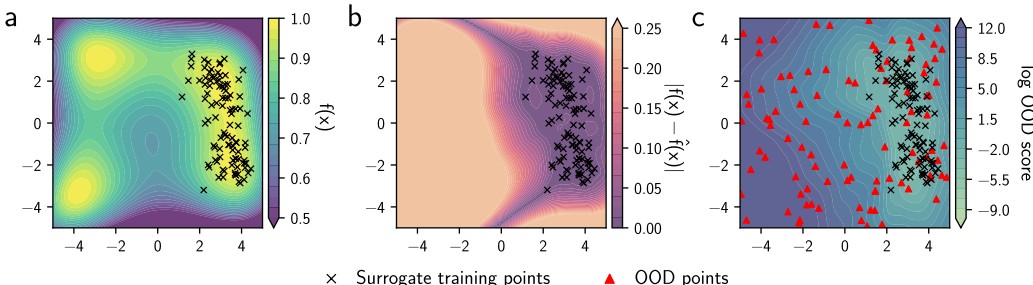

× Surrogate training points     ▲ OOD points

Figure 2: Two-dimensional test of the OOD classifier. (a) The ground truth function, $f(\boldsymbol{x})$, that we aim to estimate with a surrogate model. Scatter points indicate the training data used to fit the surrogate model. (b) Absolute error between the trained surrogate model $\hat{f}(\boldsymbol{x})$ and the ground truth function. (c) Logarithm of the OOD scores produced by an OOD classifier that was trained using the black and red scatter points as negative and positive training examples, respectively. Arrows on ends of colorbars indicate that all values in the direction of the arrow are shown as the same color.

function is commonly used for testing non-convex optimization methods because it contains multiple local optima. We illustrate a case where the training data for the surrogate model is limited to a small region of the input space mimicking real-world problems, such as in protein engineering, where data usually clusters around a natural sequence that represents a local optima. Figure 2a shows the ground truth function along with the positions of the training inputs for the surrogate model. For this example, the training data labels are the exact values of the ground truth function at the input points, with no noise added, i.e. $y_i = f(\boldsymbol{x}_i)$.

We fit a two-layer MLP to the training data using the MSE loss and the Adam optimizer (Kingma & Ba, 2015). Figure 2b shows the absolute error between the surrogate model, $\hat{f}(\boldsymbol{x})$, and the ground truth function across the input space. As expected, errors increase in regions far from the training data. In a design setting, we might optimize the surrogate model over the input space to find points with increased ground truth values relative to the points in the training set. This optimization can be effective if properly constrained to low-error regions near the training data, but might fail if the optimization can stray to other regions where the model predictions are unreliable. In offline MBO, we are unable to query the ground truth function, which means we need a method to identify trustworthy regions to constrain our search *a priori*. To identify these regions, we trained a (separate) two-layer MLP binary classifier, using the surrogate model's input training points as negative (in-distribution) training examples and points uniformly sampled across the input space as positive (OOD) examples. We then used Equation 5 to calculate OOD scores for points in the input space; these scores are shown in Figure 2c, along with the positions of the positive and negative training examples. Comparing Figs 2b and 2c, high OOD scores align with areas where surrogate model predictions deviate from the ground truth.

In this toy model, the "design distribution" is the uniform distribution over the input space, making OOD scores proportional to $1/p_{tr}(\boldsymbol{x})$. This score is a more suitable predictor surrogate model error than $s(\boldsymbol{x})$, as discussed in Section 2.2. While estimating this ratio is straightforward in a low-dimensional 2D space, it can be exceedingly difficult in high-dimensional spaces where $p_{tr}(\boldsymbol{x})$ may be arbitrarily complex. Therefore, in most practical cases, we select positive OOD examples from areas likely to be explored by a search method. These are regions of the input space that are unlikely to be densely sampled by a uniform sampling scheme, but are the most important for detecting distribution shift in designed inputs (see Appendix D for a comparison to design-independent distributions for the positive OOD class). This more focused strategy results in an OOD classifier that approximates the density ratio $s(\boldsymbol{x})$.

## 4.2 SIMULATED PROTEIN STRUCTURE DESIGN

As a sanity check before our real-world experiment, we also develop a proof-of-concept simulation scheme. We rely on protein folding prediction models (Jumper et al., 2021; Lin et al., 2023) and devise a task to optimize the folding of a small protein to its target structure, using ESMfold as a ground truth simulator. We see signs of distribution shifts caused by design, and our method can aid in selecting designed inputs by lowering regret (performance of best design vs. the performance of

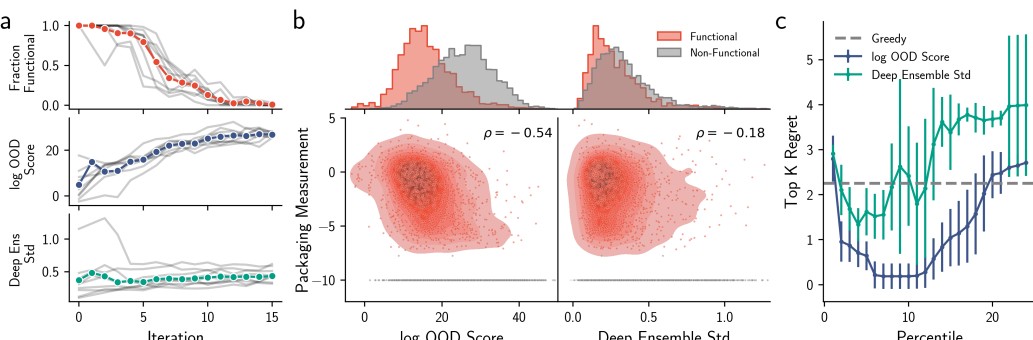

Figure 3: Application of OOD classifier to AAV engineering. (a) Experimental observation and distribution shift detection metrics over the course of design trajectories. x-axis and lines are as in Figure 1; y-axes represent the fraction of functional viruses (top), the logarithm of the OOD scores from the OOD classifier (middle) and the standard deviation of a deep ensemble (bottom) evaluated on each designed sequence. (b) Comparison of distribution shift detection metrics with experimental packaging measurement. Each scatter point represents a sequence; red points indicate functional variants and gray points indicate non-functional variants. Shaded regions are a Kernel Density Estimate (KDE) of the scatter points. Histogram (top) shows distribution of distribution shift metrics for functional and non-functional variants. (c) Evaluation of design selection using distribution shift metrics. Horizontal axis indicates the cutoff percentile of selected distribution shift detection metrics among designed sequences; Vertical axis represents the difference between the maximum observed transduction in the entire dataset and in the set of $K = 100$ sequences with the largest predicted transduction and distribution shift detection metric below the cutoff indicated by the horizontal axis. Error bars represent standard deviation estimates over 50 bootstrap resamples. Grey dashed line represents regret for $K = 100$ sequences selected greedily by predicted transduction without filtering by distribution detection metrics.

best possible design) when selecting top candidates. However, simulation settings are less relevant when real experiments can be done (Appendix F), and hence we focus the body of the paper on the latter. Interested readers can find the details in Appendix C.

### 4.3 Real-world application to the design of AAV capsid protein

We next apply our method to the design of AAV sequences. AAVs are small viruses that have been repurposed as delivery vehicles for gene therapies. Because of the complex processes involved in gene delivery, it is a real-world black box. Previous studies have demonstrated the potential of ML to improve various properties of AAVs such as manufacturability and transduction efficiency (i.e. how well the viruses can deliver genetic material to specific tissues or cell types) (Ogden et al., 2019; Bryant et al., 2021).

We consider the problem of designing AAV variants that maximize transduction in cell culture. A necessary precursor to successfully delivering genetic material is the proper folding of the viral capsid and encapsulation of the virus's genetic material, processes that we collectively refer to as "packaging". We refer to variants that do not package as "non-functional". Both packaging and transduction can be quantitatively measured experimentally using standard sequencing-based techniques (see Appendix A.2 for details).

Our aim with generating data for this problem was to overcome the issues with testing MBO methods described in Section 2.5 by inspecting mutational trajectories at many points over the course of a design procedure. We began by following a standard MBO procedure. First, we used an initial training dataset containing AAV sequence variants associated with packaging and transduction measurements to train surrogate models $\hat{f}_{\text{pkg}}(\boldsymbol{x})$ and $\hat{f}_{\text{tsd}}(\boldsymbol{x})$ to predict packaging and transduction, respectively, for a given sequence $\boldsymbol{x}$. We then used AdaLead to optimize $\hat{f}_{\text{tsd}}(\boldsymbol{x})$ under the constraint that $\hat{f}_{\text{pkg}}(\boldsymbol{x}) > \gamma$ for a chosen cutoff $\gamma$. AdaLead maintains a pool of $N$ candidate sequences and iteratively updates this pool in a greedy manner to optimize the objective function. Because genetic algorithms are local search methods, we run the optimization starting from 9 distinct starting sequences for 15 iterations each. To study variability across search methods, we also used a variant of beam search to generate

another pool of designed sequences for experimental validation; the details of this method and results associated with these sequences can be found in Appendix A.2. After the design was completed, measured packaging and transduction experimentally for all $15N$ sequences generated along each trajectory, for a total of about $5,000$ sequences. Figure 1 shows various properties of this data as a function of algorithm iteration; in particular, this figure demonstrates that a significant distribution shift occurs over the course of the design trajectories. The top panel in Figure 3a additionally shows the fraction of functional viruses generated at each step of the design. Since all designed sequences were predicted to package, the drop in the number of functional sequences demonstrates the frequency of adversarial examples.

We also evaluate the use of different model architectures including large language models (LLMs) pretrained on hundreds of millions of protein sequences adapted to our regression task with linear probing. In Appendix E, we show these models are subject to the same distribution shift that we see from training surrogate models from scratch on our data.

We tested two metrics for detecting distribution shift. The first used OOD scores outputted from an OOD classifier trained to classify the training data as negative samples and the designed sequences as positive examples. The OOD classifier has an identical architecture as the surrogate model described in Appendix A.2 except using a binary cross-entropy loss instead of MSE. The second uses the standard deviations of predictions from an ensemble of surrogate models, which we refer to as Deep Ensemble uncertainties (Lakshminarayanan et al., 2017). The middle and bottom panels of Figure 3a show the values of these metrics as a function of the optimization iteration. We can see that the OOD scores steadily increase over the course of the trajectory, in concert with the increase in surrogate model MSE shown in Figure 1b. This shows that the OOD scores can be effectively used as a continuous predictor of the intensity of distribution shift at points along a design trajectory, rather than only as a binary predictor of whether a point is in- or out-of-distribution. In contrast, the Deep Ensemble scores cannot effectively serve as a quantitative predictor of shift intensity.

Figure 3b compares the distribution shift detection metrics to packaging measurements for all designed sequences. The upper histograms compare the ability of the detection metrics to separate Functional and Non-Functional designed variants. Clearly, the OOD scores are better able to distinguish between these two categories than the Deep Ensemble uncertainties, indicating that the OOD scores can be used to determine whether a designed sequence is adversarial. Further, the lower KDE plots demonstrate that the OOD scores are a fairly reliable predictor of the continuous packaging measurement, again indicating that the OOD scores can be used as a continuous indicator of the intensity of distribution shift at a given input point.

We next evaluate using the OOD scores for selecting designed sequences, as discussed in Section 2.4. Typically, a designer will use surrogate model scores to select a small subset of designed for experimental validation. We replicate this by selecting only $K$ designed sequences out of all designs (for which in this case we know, but don't use, the ground truth). Using a cutoff scheme, sequences with the highest predicted transduction values were chosen after filtering out sequences with a distribution shift score above a certain threshold (e.g., OOD score $> 10$). We use regret as our success measure, which measures the gap between the maximum transduction value found in the full set and the $K$ selected sequences. Figure 3c displays this regret across various cutoffs and 50 bootstrap data samples. We used percentiles for the cutoffs to enable comparison between the OOD score and Deep Ensemble uncertainty. Selecting sequences with OOD scores consistently led to lower regret compared to selecting with Deep Ensemble uncertainty, even achieving zero regret in a number of cases. We note that the regret eventually increases as the cutoff is increased because more adversarial examples are included in the set of selected sequences, replacing the sequences with higher observed transduction. See Appendix A.2 for analogous regret plots at multiple settings of $K$ and for different statistics of the observed transduction values in the selected set.

## 5 DISCUSSION

Our method effectively reduces distribution shifts in sequence design by training a binary classifier to differentiate training data from designed sequences. The model's logit scores effectively identify varying distribution shift intensities. Large gaps between distributions $p$ and $q$ may result in inaccurate density ratio estimates by the OOD classifier, highlighting the need for $q$ to overlap with $p$. This can be addressed with a telescoping product (Rhodes et al., 2020), which we leave for future work. We also examine the difficulties in assessing offline MBO with static datasets. To address this, we

conducted a real-world experiment deploying offline MBO in a challenging task, focusing on the interaction between MBO and distribution shift. Our experimental results confirm that this approach successfully detects distribution shifts and improves established design approaches when used in conjunction.

Our work leaves open a number of avenues for future expansion. In particular, we prioritized demonstrating the successful application of a particularly promising and easy-to-implement method (the OOD classifier) in a real-world problem. We think the simplicity of the approach makes it a promising candidate for other domains where MBO is applied, but testing them in the real-world would require validation by practitioners on those fields.

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

## A    EXPERIMENT DETAILS

### A.1    2D TOY EXAMPLE DETAILS

Here we provide more details on the two dimensional toy example discussed in Section 4.1.

**Ground truth function** The ground truth function in this problem is given by

$$f(\boldsymbol{x}) = -\frac{1}{m}\mathrm{himm}(\boldsymbol{x}) + 1 \tag{6}$$

where $m = \max_{\boldsymbol{x} \in \mathcal{X}} \mathrm{himm}(\boldsymbol{x})$, with $\mathcal{X} = [-5, 5] \times [-5, 5]$, and $\mathrm{himm}(\boldsymbol{x})$ is the Himmelblau function, given by

$$\mathrm{himm}(\boldsymbol{x}) = (x_0^2 + x_1 - 11)^2 + (x_0 + x_1^2 - 7)^2. \tag{7}$$

The modifications to the Himmelblau function in Equation 6 have the dual purpose of (i) negating the function so that the local optima are maxima, which is aligned with the formulation of MBO in Section 2.2 as a maximization problem and (ii) normalizing the function so that all ground truth values are between 0 and 1 in the input space, which enables stable model training.

**Training data for surrogate model** The training data for the surrogate model was generated by randomly selecting points in the input space according to the distribution

$$p(\boldsymbol{x}) \propto \begin{cases} \exp(25 \cdot f(\boldsymbol{x})) & \text{if } x_0 > 0 \\ 0 & \text{otherwise.} \end{cases} \tag{8}$$

Specifically, we created a grid with width 0.005 in each direction over the input space, evaluated each point according to $p(\boldsymbol{x})$, normalized the probabilities, and sampled 100 points from the grid according to the probabilities. This distribution has the effect of concentrating the training data around the local maxima of the ground truth function in the region $x_0 > 0$. The labels for the sampled training points are simply noiseless evaluations of the ground truth function at those points.

**Surrogate model details** The surrogate model for this example was an MLP with two fully connected hidden layers with 200 nodes each and ReLU activation functions applied to each hidden layer. The output of the model was scaled using a one-dimensional Batch Normalization. The model was trained by minimizing an MSE loss over 1000 epochs using the Adam algorithm a batch size of 32, an initial learning rate of 1e-3, and a dropout rate of 0.1 for hidden layer parameters.

**OOD classifier details** The training data for the OOD classifier consisted of (i) the input training points for the surrogate model, associated with a label of 0 representing the "in-distribution" class, and (ii) 100 input points sampled uniformly at random from the input space associated with a label of 1 representing the OOD class. The architecture of the OOD classifier was identical to that of the surrogate model; the only change being the removal of the final batch normalization layer and the addition of L2 regularization over the weights with regularization strength 1e-3, to prevent overfitting. The model was trained by minimizing the binary cross-entropy loss over 1000 epochs using Adam algorithm a batch size of 32, and an initial learning rate of 1e-3, and a dropout rate of 0.1 for hidden layer parameters.

### A.2    AAV EXPERIMENT DETAILS

**Data generation** Both the initial training data and designed data contain AAV variants associated with transduction and packaging measurements. In both datasets, the sequences are variants of the AAV9 wild-type sequence, modified in a 63 amino-acid region containing the VR-IV loop of the VP3 capsid protein. The distribution of edit distances to AAV9 WT for the training and design datasets are shown in Figure 4.

Sequences in the training and designed data were assayed for packaging and transduction using a standard sequencing-based technique Ogden et al. (2019). This technique involves first constructing a "library" of plasmid sequences that encode the protein variants of interest. This plasmid library is then subjected to experimental conditions that enable the plasmids to be converted to proteins and assemble into viral capsids, to produce a sample of viruses containing genetic material. This

virus sample is then introduced to cells, allowing the viruses to enter these cells and transfer their genetic material to the cell's nucleus if they are capable of doing so; this is the process known as transduction. The genetic material that successfully entered the nucleus of cells is then collected. At each stage of the experiment, a small sample of genetic material is sequenced using Next Generation Sequencing methods, allowing one to approximate the abundance of each sequence variant in the plasmid library, the virus sample, and the sample of successfully transduced genetic material. These abundance measurements are in the form of sequencing counts, i.e. the number of times a specific variant appears in the sequencing data. Let $n_i^{\text{plasmid}}$, $n_i^{\text{virus}}$ and $n_i^{\text{tsd}}$ be the sequencing counts of the $i^{th}$ variant in the plasmid, viral and transduced samples. The ability of variant $i$ to package is then quantified as the log rate:

$$y_i^{\text{pkg}} = \log \frac{n_i^{\text{virus}}}{n_i^{\text{plasmid}}} \tag{9}$$

Similarly, the transduction ability of variant $i$ is quantified as

$$y_i^{\text{tsd}} = \log \frac{n_i^{\text{tsd}}}{n_i^{\text{virus}}}. \tag{10}$$

**Design strategy** We run offline MBO given a fixed training dataset of $(\boldsymbol{x}_i, y_i^{\text{pkg}}, y_i^{\text{tsd}})$ triplets, where $\boldsymbol{x}_i$ is the sequence of variant $i$. The input space for the design is the space of all amino acid sequences of length 63. Details of the surrogate models and search method used in this MBO design are discussed in turn below.

**Surrogate models** We trained two surrogate models: $\hat{f}_{\text{pkg}}(\boldsymbol{x})$ to predict packaging from sequence and $\hat{f}_{\text{tsd}}(\boldsymbol{x})$ to predict transduction. These packaging and transduction surrogate models were trained using the input-label pairs $(\boldsymbol{x}_i, y_i^{\text{pkg}})$ and $(\boldsymbol{x}_i, y_i^{\text{tsd}})$, respectively. Both models had a Convolutional Neural Network (CNN) architecture with following hyperparameters:

- Number of Convolutional Blocks: 2
- Number of Channels: $[32, 32]$
- Pooling Scales: $[0, 0]$
- Number of dense layers: 1
- Dense layer size: 32
- activation type: LeakyReLU

Both models were trained by minimizing an MSE loss using the Adam optimization algorithm until convergence using early stopping on a random-holdout validation set (10% of samples) with a patience set to 10 epochs.

We used an identical architecture for training the OOD classifier except we used a binary cross-entropy loss instead of a mean squared error (MSE).

**Search algorithm**. We applied two search methods to this problem. The first is AdaLead (Sinai et al., 2020), a variant of a genetic algorithm that has been shown to work well for biological sequence design applications. We refer readers to Sinai et al. (2020) for a detailed treatment of the AdaLead algorithm. The second search method we applied is a stochastic variant of beam search, with a beam width of 5, maximum number of iterations equal to 15 and a total budget of 5,000.

**Results**. Results for sequences designed with AdaLead are shown in Figure 1 and 3 in the main text. Analagous results for the sequences designed with beam search are shown in Figures 5 and 6, below. We also report additional results for the selection regret experiment shown in Figures 3c and 6. In particular, we show the regret as we vary the number of selected sequences ($K = 10, 50, 100, 250$) and the batch statistic for the selected sequences (90th percentile, 95th percentile, and Max). We show this for the AdaLead design in Figure 7 and the beam search design in Figure 8.

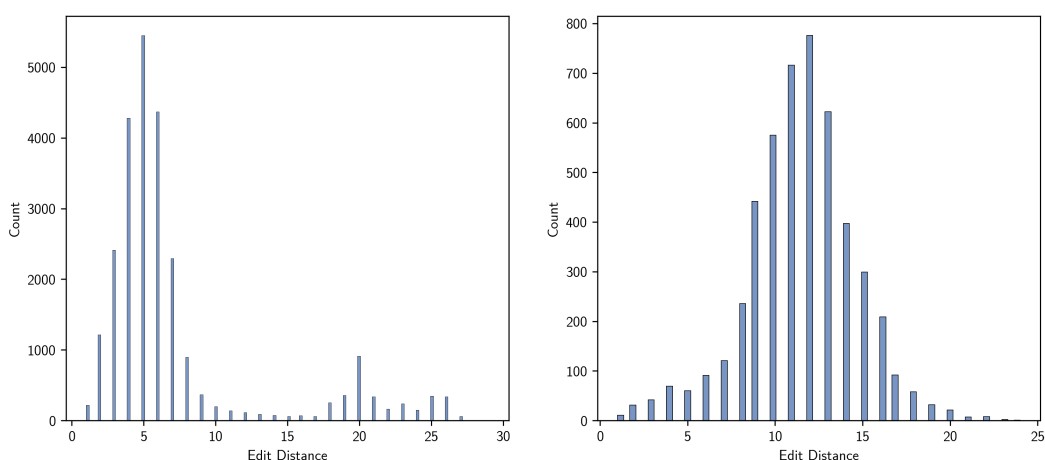

Figure 4: Distribution of edit distances to wild-type for training data (left) and designed data (right).

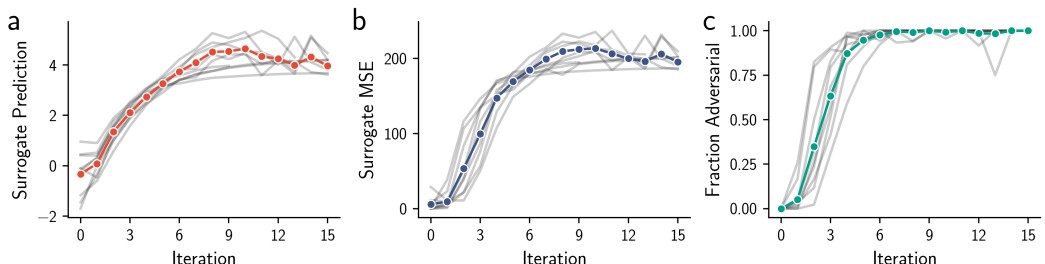

Figure 5: Analysis of distribution shift in AAV variants designed by MBO with beam search as the search method. Plot descriptions are as in Figure 1 in the main text.

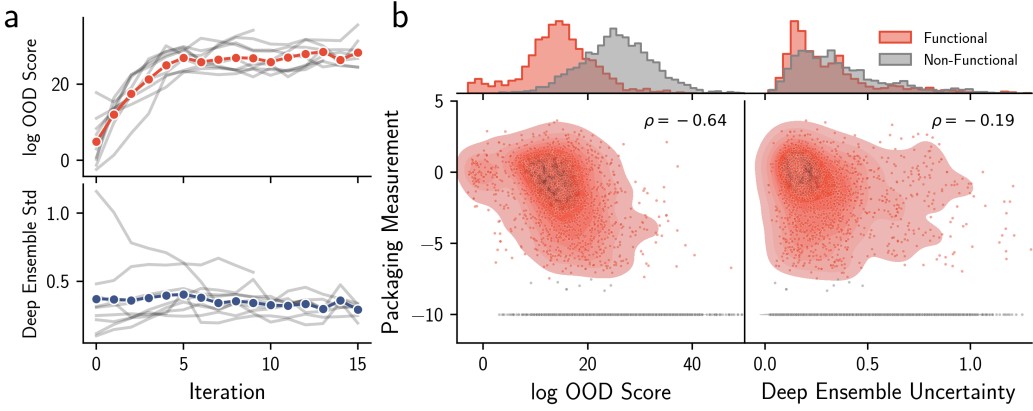

Figure 6: Results applying distribution shift detection metrics to AAV variants designed by MBO with beam search as the search method. Plot descriptions are as in Figure 3 in the main text.

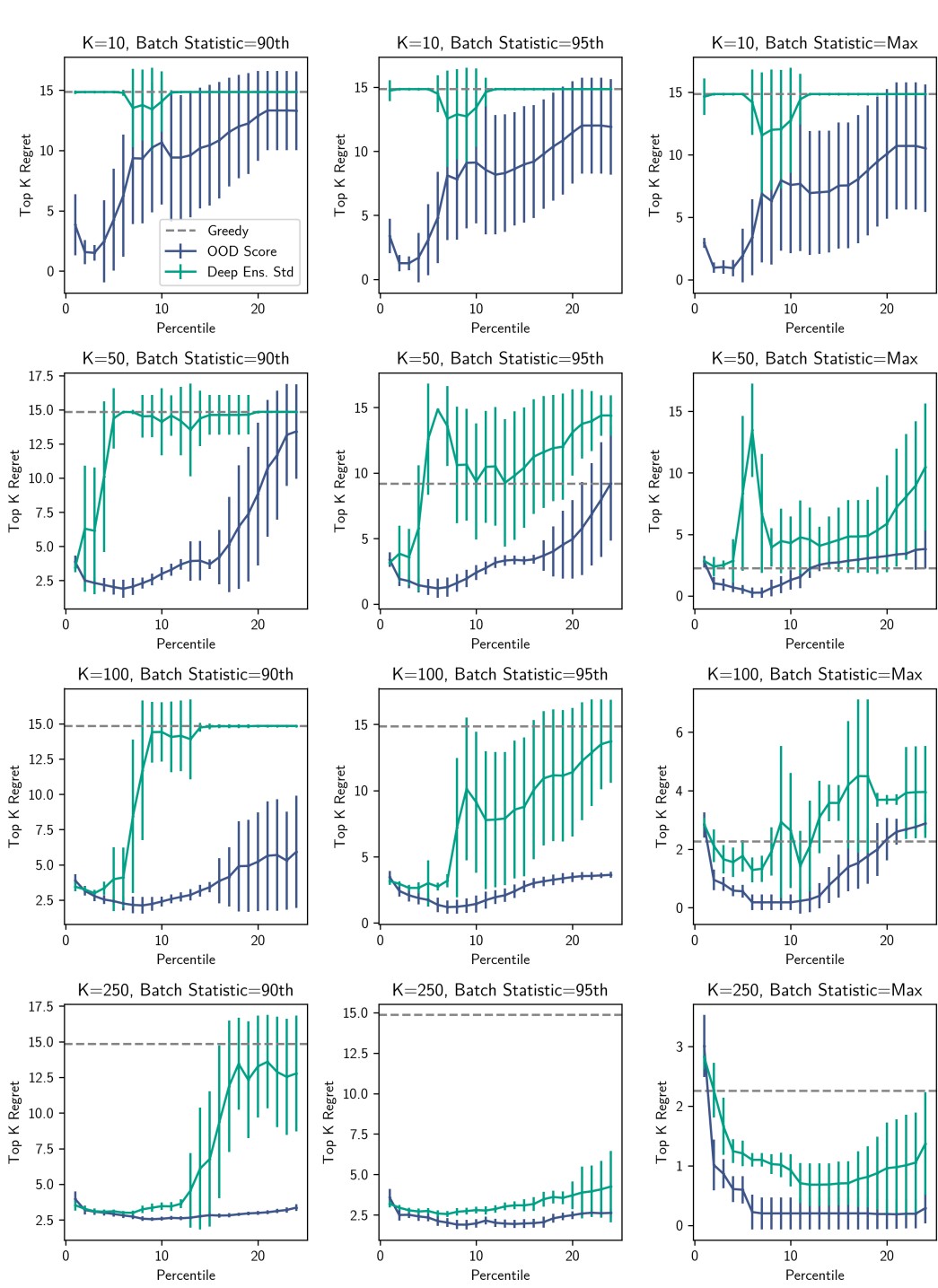

Figure 7: Expanded results of selection regret experiments for AAV variants designed by MBO with AdaLead as the search method. Regret is shown for varying batch sizes (Top K) and batch statistics (90th, 95th, Max). Plot descriptions are as in Figure 3c.

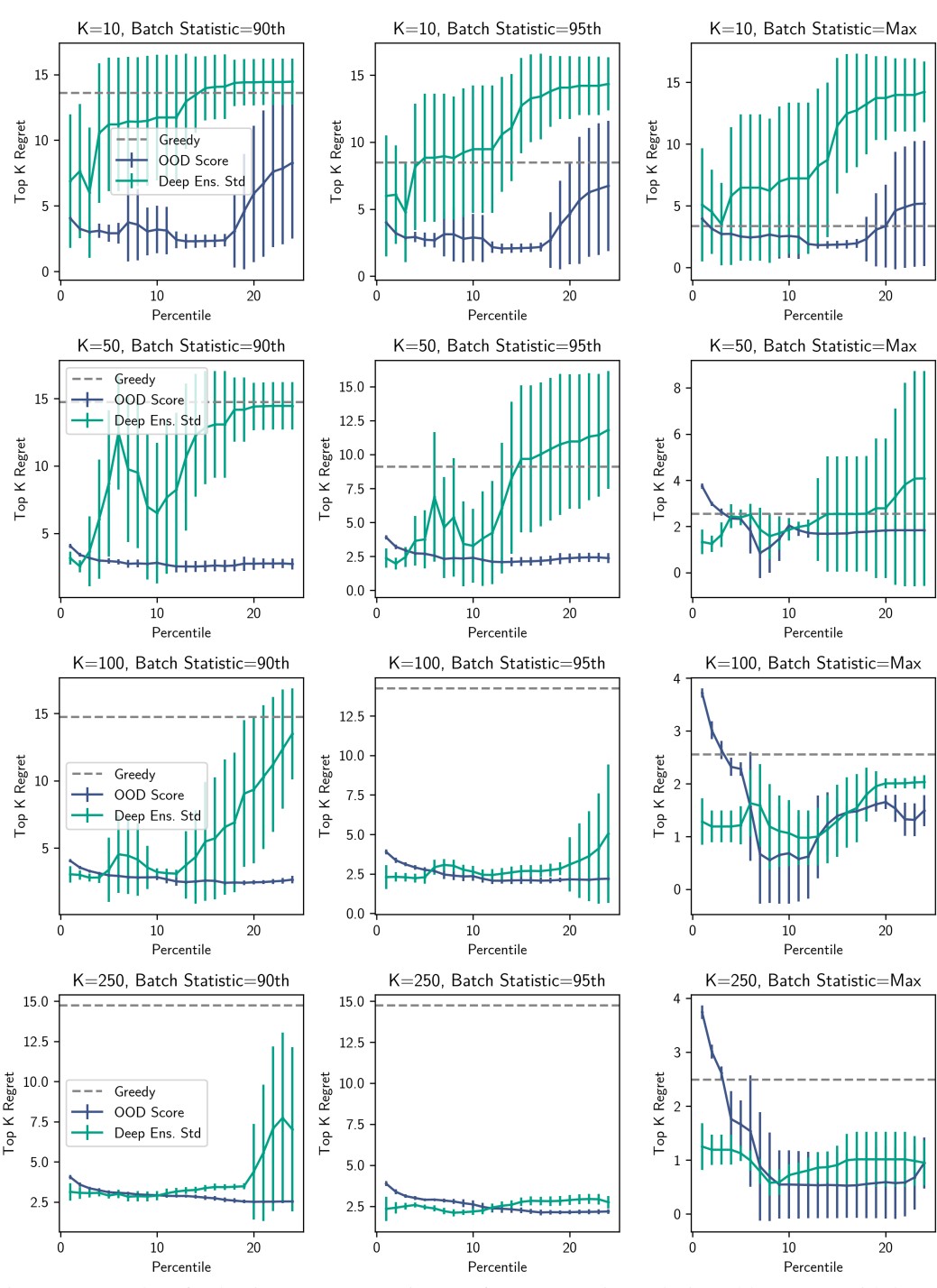

Figure 8: Results of selection regret experiments for AAV variants designed by MBO with beam search as the search method. Regret is shown for varying batch sizes (Top K) and batch statistics (90th, 95th, Max). Plot descriptions are as in Figure 3c

## B    BALANCE OF TERMS IN OOD SCORES

In Section 2.2, we discuss the distinction between the OOD score, $s(\boldsymbol{x}) = p_{\text{de}}(\boldsymbol{x})/p_{\text{tr}}(\boldsymbol{x})$, and another potential distribution shift detection metric, $1/p_{\text{tr}}(\boldsymbol{x})$. We suggest that the latter may be a more suitable score for detecting distribution shift that would result in surrogate model error, but use the OOD score instead because it is straightforward to calculate via the density ratio trick. Further, we claim that the difference in OOD scores between two nearby design points will tend to be dominated by the denominator terms in the OOD scores, and thus the distinction between $s(\boldsymbol{x})$ and $1/p_{\text{tr}}(\boldsymbol{x})$ will often be negligible in practice. Here, we provide justification for this latter claim by considering a simple one dimensional example. In particular, consider an input point $x$ drawn from a one dimensional design distribution with continuous support, $p_{\text{de}}(x)$, and a nearby point $x + \delta x$. The difference between the log OOD scores of these two points can be approximated by Taylor expanding to first-order:

$$\log s(x + \delta x) - \log s(x) \approx \delta x \frac{d}{dx} \log s(x), \tag{11}$$

which can then be simplified by computing

$$\frac{d}{dx} \log s(x) = \frac{p'_{\text{de}}(x)}{p_{\text{de}}(x)} - \frac{p'_{\text{tr}}(x)}{p_{\text{tr}}(x)}, \tag{12}$$

where prime indicates differentiation with respect to $x$. Now we assume that both the design and training distribution are Gaussian distributions, with means $\mu_{\text{de}}$ and $\mu_{\text{tr}}$, respectively, and variances $\sigma_{\text{de}}^2$ and $\sigma_{\text{tr}}^2$, respectively. In this case, the difference between the scores is approximately

$$\log s(x + \delta x) - \log s(x) \approx \frac{x - \mu_{\text{tr}}}{\sigma_{\text{tr}}^2} - \frac{x - \mu_{\text{de}}}{\sigma_{\text{de}}^2} \tag{13}$$

We now consider cases where the first term in Equation 13, which corresponds to the $1/p_{\text{tr}}(\boldsymbol{x})$ term in the OOD score, will be larger than the second term. If the variances of the train and design distributions are roughly similar, then the first term will tend to be larger because $x$ is drawn from the design distribution and is therefore likely to be closer to the mean of the design than the training distribution, as long as these means are well-separated. The impact of the second term will be further reduced if the variance of the design distribution is large compared to that of the train distribution. To summarize, the two conditions that will cause the $1/p_{\text{tr}}(\boldsymbol{x})$ term to dominate the difference in OOD scores between two nearby points are (1) the means of train and design distribution are well separated and (2) the variance of the design distribution is larger than that of the train distribution.

In practice, both of these conditions are typically satisfied by design distributions. The first condition is satisfied because the design method will tend to search around regions of input space far from the training data (thus inducing the distribution shift that is the focus of this paper. Similarly, the second condition is usually satisfied because the design method will tend to search large regions of the input space to find candidate solutions, producing a large variance in the design distribution. Thus, we can usually assume that the difference in $1/p_{\text{tr}}(\boldsymbol{x})$ terms is the dominant effect when considering the difference in OOD scores between two input points.

## C    PROTEIN STRUCTURE PREDICTION EXPERIMENT

**Problem Description**. Real-world sequence design problems are characterized by high-dimensional discrete input spaces, label noise, and limited training data. While evaluating design methods with physical experiments in the sciences and engineering is the best way to evaluate offline MBO, these experiments can be resource and time-intensive. Therefore, it is valuable to have simulation settings that mimic key aspects of the target problem in order to develop and evaluate methods rapidly (Trabucco et al., 2022). Here we describe a simulation using protein structure prediction (Jumper et al., 2021) as a benchmark for evaluating offline MBO. The challenge in protein structure prediction is to determine the 3D shape of a protein based solely on its amino acid sequence. Knowing this 3D shape is key to understanding how the protein functions and interacts on a molecular level. Deep learning has recently brought about significant advances to the field. AlphaFold2 (Jumper et al., 2021) has achieved impressive accuracy in predicting protein shapes, with some predictions reaching the accuracy levels of experimental methods.

Our proposal is to use a protein structure prediction network as the ground truth function given its broad generalization capabilities across a wide variety of proteins. Concretely, this means our task is to design an amino acid sequence that folds into the ground truth structure for a predefined protein. The ground truth structure is a publicly available experimentally derived structure (e.g., by using X-ray crystallography). Because there are many amino acid sequences that can fold into the same structure, this is a design task to search in the amino acid sequence space for a sequence that has a predicted structure with the minimum structural distance to the target structure. The wild-type sequence that encodes this structure is intentionally hidden from the model.

Using a protein structure prediction network as a simulated fitness landscape offers several benefits. These networks provide accurate results across various protein types and sequence lengths. They are computationally efficient to query, free from label noise, and allow us to examine performance differences across multiple datasets. Most importantly, we observe distribution shift induced by design, which we find is a salient feature of real-world protein engineering settings.

Our aim is to design a protein sequence that folds into the structure of Trp-Cage (Zhou, 2003), a notably compact 20-residue mini protein known for its stable folding and structural elements. To determine how closely the predicted structure of our designed sequence matches the true structure of Trp-Cage, we employ the frame-aligned point error (FAPE) metric (Jumper et al., 2021). Lower FAPE scores indicate a closer alignment between the predicted and the actual structure of Trp-Cage.

**Experimental Design**. We design a training dataset by employing an in-silico variant of error-prone PCR (Smith, 1985), a lab method that randomly mutates a system by decreasing the precision of DNA replication. Here, we introduce mutations at each position in the wild-type (WT) protein sequence (i.e., an example protein sequence that is known to fold into the target protein structure), based on a specific error rate $\epsilon$. We loop over every position in a sequence and with probability $\epsilon$ we mutate each position using a uniform distribution over the 20 canonical amino acids. This process is repeated 10K times to generate a set of unlabeled sequences. In our setting we set $\epsilon$ to 0.5, so roughly half of the positions of the wild-type sequence are mutated. This makes the problem sufficiently challenging with a lot of diversity in the sequences (and subsequently the distances between their predicted structures and the ground truth). For each sequence, we compute predicted structures using ESMFold (Lin et al., 2023) and FAPE scores by comparing each predicted structure to the WT's known crystal structure. Negative FAPE scores are used as the labels for our dataset since our aim is to minimize the structural distance to the target protein (or maximize the negative FAPE score).

We implement offline MBO with this training dataset. We first fit a surrogate regression model $f(\boldsymbol{x})$ to the $(\boldsymbol{x}_i, y)$ pairs, predicting FAPE scores from the amino acid sequence. Then, we design 10K sequences using the same genetic algorithm used in the AAV experiment ((Sinai et al., 2020) to maximize $f(\boldsymbol{x})$. Because genetic algorithms are local search methods, we run the optimization from 10 different starting points corresponding to 10 distinct initial sequences, restart each optimization run with two random initializations, and save designed sequences from iterations 5, 20, 50 corresponding to low, medium, and high shift intensities. After running optimization, we train the OOD classifier to separate the training data (class 0) from the designed data (class 1). We then label our data by using our ground truth function (ESMFold) to compute FAPE scores, measuring the structural distance between each amino acid sequence and our target protein structure.

**Design hyperparameters** We use an identical set of hyperparameters to what is described in Appendix A.2 for the AAV experiment for the surrogate regression model. For search, we use Adalead (Sinai et al., 2020) as our optimization algorithm with a population size of 1K. We refer readers to Sinai et al. (2020) on algorithm implementation and recommended hyperparameters.

**Results**. We first show evidence that our simulation shows a distribution shift between the design data and the training data (Fig. 9a and in a more extensive ablation in Fig. 10). Here we evaluate the regression model accuracy (as measured by Spearman Rank Correlation) across optimization steps (holding edit distance fixed to 10) and across edit distances (holding optimization steps fixed at 20). We observe substantial distribution shifts in both settings corresponding to feedback and non-feedback covariate shift.

Next, we show how a distribution shift metric can enable better selection of variants. We follow an identical setup to the AAV experiment described in Section 4.3 and show top K regret for a variety of values of $K$ and batch statistics (90th, 95th, Max) (Fig. 11). In all cases we see the OOD score achieves lower regret than Deep Ensemble Uncertainty.

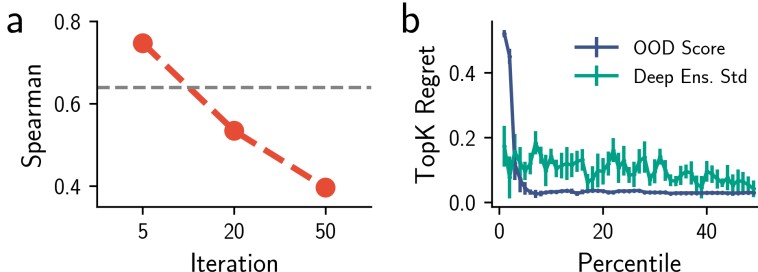

Figure 9: Application of OOD classifier to a protein folding simulation task using a protein structure prediction network as the ground truth oracle. The goal is to design a sequence that folds into a target 3D protein structure. Offline MBO is run on a synthetically generated dataset. (a) Model accuracy along an optimization trajectory. (b) Evaluation of design selection using distribution shift detection metrics.

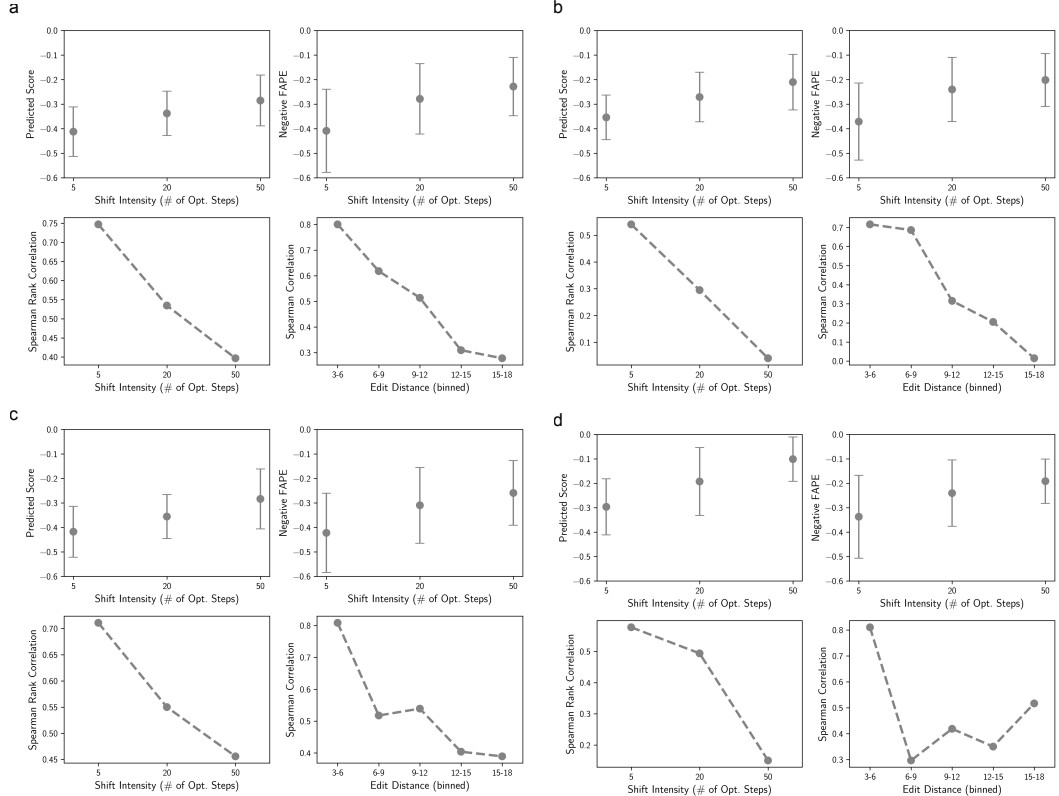

Figure 10: (Top) Predicted score (left) and ground truth measurement (right) increase as a function of optimization step. (Bottom) Model performance decays as a function of optimization step (left) and edit distance (right). Subpanels correspond to four randomly sampled datasets given the parameters to the in-silico error-prone PCR generation procedure.

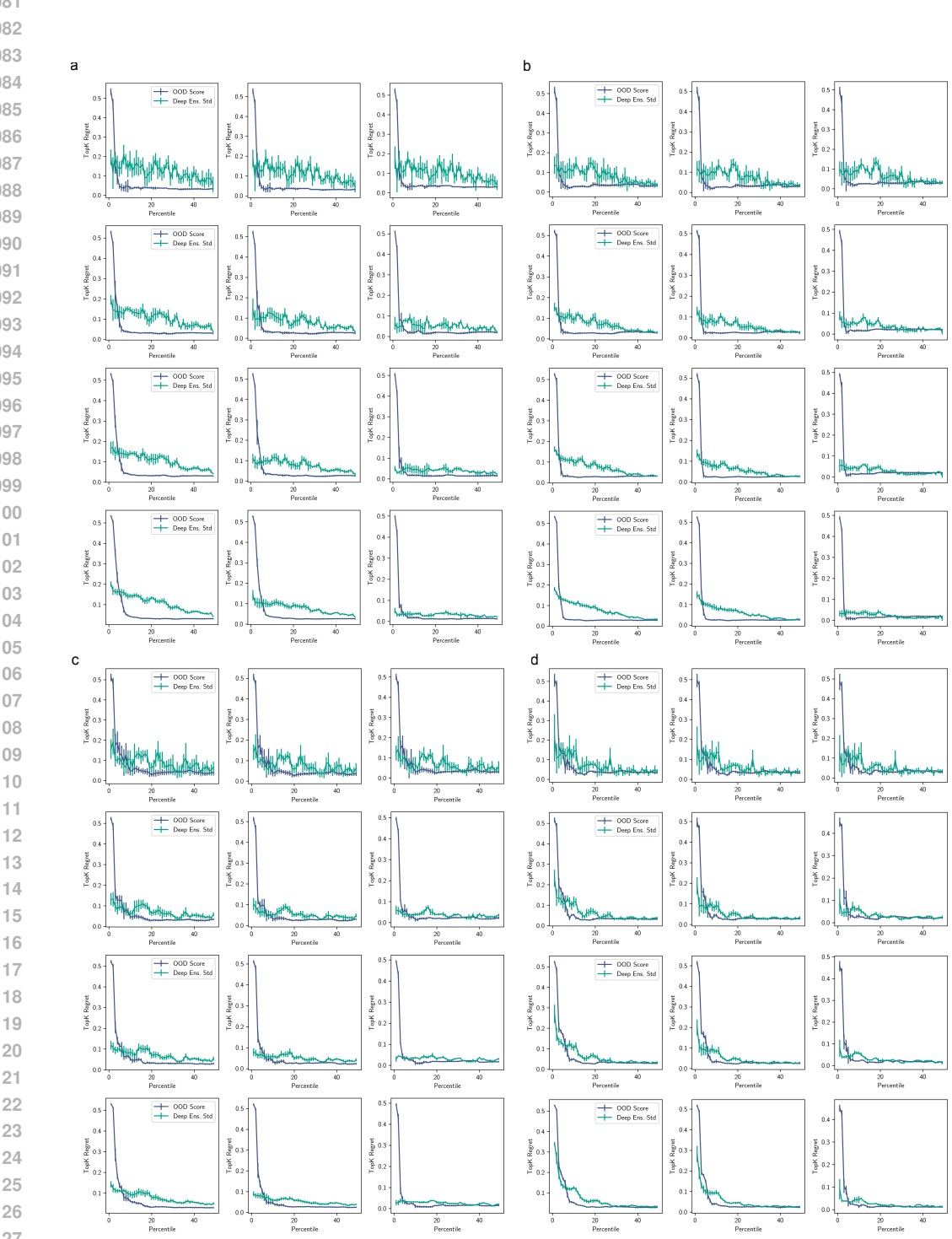

Figure 11: Regret plots for the protein structure prediction task for varying batch sizes and batch statistics. Rows are Top K = 10, 50, 100, 250 and columns are transduction percentiles 90th, 95th, Max. Subpanels correspond to four randomly sampled datasets given the parameters to the in-silico error-prone PCR generation procedure.

# D    CHOICE OF Q

We find that the use of a design-induced test set, rather than a fixed test, as the q distribution is crucial to the performance of our method. To our knowledge, this choice of q distribution is not discussed in the density ratio estimation literature, which typically assumes access to a fixed test set. Here we evaluate three choices of a q distribution: (1) Uniform distribution over the input domain, (2) 1-15 random mutations to the wildtype background sequence, and the (3) Design set.

We evaluate these three methods in the context of the AAV dataset described in Figure 3. For (1), we fix the wildtype context and draw a sequence of length 63 (the modifiable region) from a uniform distribution. For (2), we start with the wildtype sequence and perform ancestral sampling to draw sequences: randomly select number of mutations from a uniform distribution over 1-15 mutations, then randomly select positions to modify and then draw a mutation from a uniform distribution over AAs. For (3) we use the method described in the main text, which uses the designed sequences as the q distribution.

We compare the three choices by evaluating how well the scores serve as a predictor of whether a designed variant is functional by looking at the receiver operating characteristic (ROC) curve and the corresponding area under curve (AUC) score and find substantial improvements in detecting functional variants using the design distribution (in blue) over the uniform distribution (in green) and the wild-type conditoined uniform sampling strategy (in red).

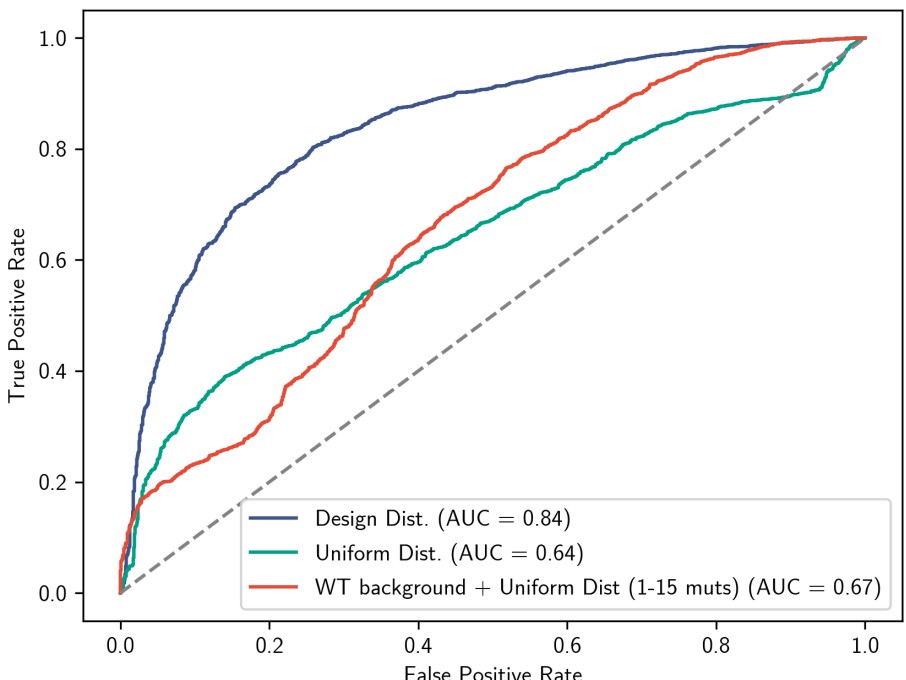

Figure 12: Comparing three choices of a q distribution: (1) Uniform distribution over the input domain, (2) 1-15 random mutations to the wildtype background sequence, and the (3) Design set. We evaluate the three choices on the AAV experiment in Figure 3 by looking at the discriminative capabilities of the score to separate designed variants that are functional versus non-functional

# E  DO PRETRAINED LLMS ADDRESS FEEDBACK COVARIATE SHIFT?

It has previously been shown that model pretraining can improve robustness to distribution shift (Hendrycks et al., 2019). In the protein domain, large language models (LLMs) have been pretrained on hundreds of millions of protein sequences (Elnaggar et al., 2021; Lin et al., 2023). These models have shown impressive performance on a variety of downstream prediction tasks. Here we show that pretrained LLMs do not address feedback covariate shift in the design setting. We use ProtBERT (Elnaggar et al., 2021), a pretrained language model, to predict our target property using linear probing (Kumar et al., 2022), a method for adapting pretrained models to downstream regression problems by fixing the pretrained weights and fitting a linear head to predict the target property. The linear head is fit to the average of the per-token embeddings. We used an identical train/test split to the CNN in Figure 3 and computed predictions on the designed sequences. We observe a substantial drop in predictive performance from a random holdout (Spearman rho = 0.75) to the design set (Spearman rho = -0.55). In Fig Xa, we observe LLM model scores increase as a function of iteration, which maps to an increase in the surrogate error (measured by MSE) in Fig Xb. Fig Xc shows the fraction of non-functional designs, also increasing as a function of iteration.

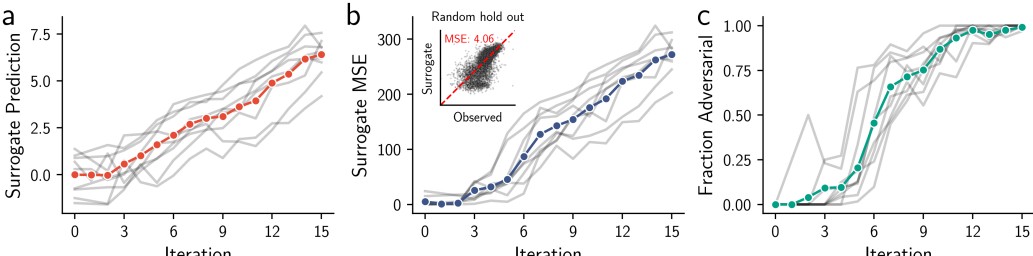

Figure 13: Pretrained LLMs do not address feedback covariate shift. (a) Predicted property value from the surrogate model. (b) MSE between surrogate predictions and observed experimental measurements of the property. Inset shows surrogate predictions versus observed property values for a randomly held out set of data from the training distribution. (c) Fraction of sequences observed to be adversarial examples.

## F  FEEDBACK COVARIATE SHIFT IN SIMULATION BENCHMARKS

Demonstrating feedback covariate shift in simulated environments like Design-Bench (Trabucco et al., 2022) or FLEXS (Sinai et al., 2020) has proven to be challenging. This is supported by the unexpectedly high performance of unconstrained maximization algorithms, such has the $max_{\boldsymbol{x}} f(\boldsymbol{x})$ method in Design-Bench and the performance of Adalead, a genetic algorithm used to implement maximization in a discrete search space, in FLEXS. Running this method out-of-the-box in a real-world setting can produce a 0% functionality rate as we show in our Figure 3 experiment. To provide additional support for this, we ran a simulation in FLEXS using Adalead with an identical set of hyperparameters that we used to design the AAV experiment and we find a minimal drop in predictive performance between a random holdout (Spearman $\rho = 0.61$) and the design set (Spearman $\rho$=0.49), 88.5% of the designed sequences were functional, and the above histogram clearly shows the design algoritm successfully designed a batch of sequences (in red) that have a higher distribution of measurements compared to the training data. The strong performance of the naive maximization baseline in a variety of datasets covered in the simulation benchmarks coupled with our experimental results providing additional confirmatory evidence of this result, suggests that simulation settings do not effectively mimic the feedback covariate shift setting found in real-world data problems.

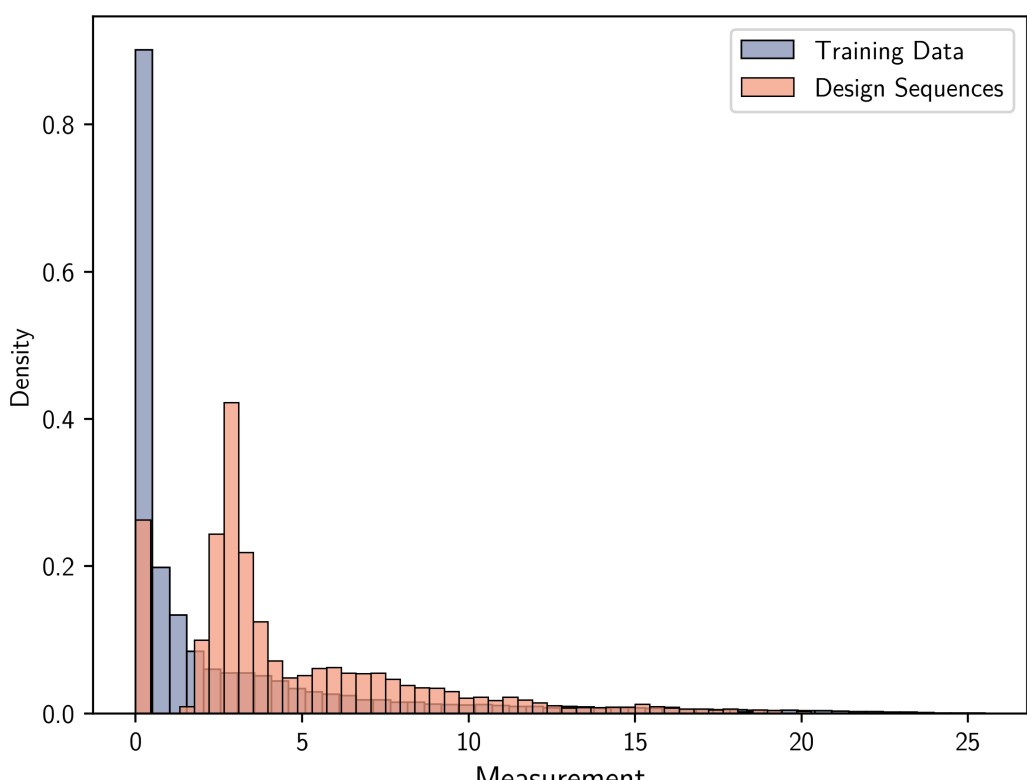

Figure 14: Optimization of a RNA binding landscape using Adalead, a genetic algorithm that maximizes a reward function. The algorithm successfully optimizes this commonly used RNA binding landscape in the FLEXS benchmark as evidenced by the shift in the ground truth measurements for the designed sequences compared to the training data.

