# OpenReview forum: "Improving real-world sequence design with a simple meta-heuristic for detecting distribution shift"
_ICLR.cc/2025/Conference — Submitted to ICLR 2025_

### Official Review · Reviewer_9umf · 2024-11-03

**Soundness:** 4
**Presentation:** 3
**Contribution:** 3
**Rating:** 8
**Confidence:** 4

**Summary:**

This paper presents a method for detecting distribution shifts in machine learning-guided biological sequence design design, specifically addressing model-based optimization (MBO) prediction reliability when exploring regions distant from training data, identifying distribution shift when it occurs. The work introduces:

1. A binary classifier approach to detect out-of-distribution samples in MBO
2. Empirical validation through AAV capsid engineering experiments
3. Comparison between simulation benchmarks and real-world distribution shift severity
4. A framework for identifying unreliable predictions during sequence optimization

**Strengths:**

1. Validation through wet-lab experiments, extending beyond simulation-based evaluation (which itself was also thorough)
2. Straightforward implementation of the proposed method
3. Demonstration that simulation benchmarks may not capture real-world distribution shift challenges
4. Comprehensive ablation studies and baseline comparisons
5. Technical foundation in density ratio estimation literature
6. Direct applicability to real-world MBO applications in biological sequence design

**Weaknesses:**

1. Limited analysis of predictor architecture and training choices' effects on distribution shift detection. Understanding the method's robustness across different model choices would strengthen the results.
2. While the method effectively identifies OOD samples, the paper provides limited guidance on what to do with these flagged sequences beyond excluding them. The practical impact would be enhanced by discussing mitigation strategies such as active learning, model retraining, or ways to incorporate OOD scores into exploration.
3. The theoretical foundations could benefit from deeper analysis, particularly regarding how classifier architecture affects density ratio estimation accuracy and potential bounds on detection performance under various distribution shift scenarios.

**Questions:**

1. Have you explored strategies for using the OOD scores beyond binary accept/reject decisions? Could the scores be incorporated into the optimization objective to guide exploration?
2. How sensitive is the method to surrogate model architecture and training procedure choice? Additional experiments testing robustness across architectures would be informative.
3. Could you elaborate on potential approaches to leverage the OOD scores for active learning or model refinement when significant distribution shift is detected?

---

> ### Author Response · Authors · 2024-11-22
>
> We thank the reviewer for their thoughtful and positive review. We respond to their questions below:
>
> > Have you explored strategies for using the OOD scores beyond binary accept/reject decisions? Could the scores be incorporated into the optimization objective to guide exploration?
>
> We have not currently explored such ways of using the OOD scores, though we agree that this is a fruitful area for future research. One challenge is that the current version of the OOD classifier requires the set of designed sequences in order to be trained, and thus it could not be used during the process of optimization without modifications. We will highlight in the Discussion section that it would be an exciting area of research to determine a reasonable method for incorporating the OOD scores into the optimization procedure itself.
>
> > How sensitive is the method to surrogate model architecture and training procedure choice?  Additional experiments testing robustness across architectures would be informative.
>
> We assume the reviewer is referring to modifications to the architecture and training procedure of the OOD classifier, as the surrogate model used in the design procedure is outside of the scope of this paper. We performed a basic hyperparameter search on the OOD classifier architecture, using binary cross entropy on a validation set as the target property. We found that architectures with fewer parameters underfit the data (i.e. did not achieve as low a validation loss), while larger models did not meaningful improve performance. We will clarify in the Appendix that we performed this basic search and that this should be done for any new application (which is straightforward, as it follows the best practices for training any binary classifier)
>
> > Could you elaborate on potential approaches to leverage the OOD scores for active learning or model refinement when significant distribution shift is detected?
>
> To clarify, the OOD scores are able to detect whether individual sequences are OOD, but are not able to distinguish between different degrees of distribution shift across the entire design set (i.e. they can not be used to say that one complete set of designed sequences is more or less OOD than another). If this is a goal, a distributional statistic such as Maximum Mean Discrepancy [1] may be more appropriate. Nonetheless, it is reasonable to ask whether the OOD scores can be used to correct for distribution shift or guide sampling in an active learning setting. Logistic regression of a similar type to the OOD classifier has been used to produce approximate importance weights in Importance Weighted Empirical Risk Minimization (IWERM) [2] to improve the performance of a regression model on the test set. It is not immediately clear how such a correction would be used in our setting given that our goal is not to improve regression performance. However, investigating how using the OOD classifier to modify the surrogate model in this way effects the prediction performance of the model is an intriguing possibility that we will mention in the Discussion section.
>
> [1] Gretton, A., Borgwardt, K. M., Rasch, M. J., Schölkopf, B., & Smola, A. A kernel two-sample test. Journal of Machine Learning Research, 13 (2012).
> [2] Shimodaira, H. Improving predictive inference under covariate shift by weighting the log-likelihood function. J. Stat. Plan. Inference 90, 227–244 (2000).

---

### Official Review · Reviewer_okPN · 2024-11-04

**Soundness:** 1
**Presentation:** 2
**Contribution:** 2
**Rating:** 3
**Confidence:** 4

**Summary:**

This paper introduces a binary classification model for detecting out-of-distribution (OOD) samples in the context of offline model-based optimization (MBO). The classification model is trained on a given offline dataset (labeled 0, in-distribution) and a generated dataset (labeled 1, OOD), where the generation algorithm is task-dependent. The learned classification model is used to calculate a score indicating the intensity of the distribution shift. Experiments were conducted in a synthetic problem, a simulated protein structure design, and a real-world Adeno-Associated Virus (AAV) capsid sequence design.

**Strengths:**

1. The proposed algorithm is straightforward and also easy to implement.
2. The paper was generally easy to understand and follow.
3. The method achieved better results in OOD detection than the uncertainty-based method (deep-ensemble).

**Weaknesses:**

1. I'm not fully convinced by the proposed method. The paper says in line 921 that $1/p_{tr}(x)$ (where $p_{tr}$ is training distribution) is more suitable for detecting distribution shift. If so, there are several ways to achieve this, e.g., kernel density estimation (KDE) or neural autoregressive density estimation (NADE; [1]). A careful conceptual and experimental comparison with these density estimation methods seems crucial.
2. I'm uncertain how the proposed method "can significantly improve design quality" (line 27). Choosing the right threshold seems critical to effectively balance the exploitation of the surrogate model and the OOD robustness. Moreover, I suspect that the optimal threshold for achieving the best design, regardless of whether a score-based or percentile-based method is used, will vary depending on the specific design task and the distribution of the training dataset. However, there has been limited discussion on threshold selection.
3. Similarly, there is no experimental evidence showing that the proposed algorithm can actually improve design quality. The experiments were only about the ability to detect OOD, specifically in comparison to deep ensembles.
4. The proposed algorithm relies on the MBO algorithm to generate the OOD dataset for classifier training. However, this paper only validates it using a single MBO algorithm, AdaLead, which raises concerns about the method’s versatility and generalizability.
5. A minor point, but the writing could be improved for better readability. For instance, it might be helpful to create a separate 'Preliminaries' section for the content in Sections 2.1 (offline MBO) and 2.2 (distribution shift), allowing the 'Method' section to focus solely on the main contribution. Additionally, the paper is somewhat verbose, particularly in the experiment section. A more concise presentation that highlights the main contributions and insights would strengthen the overall readability.

[1] Uria, Benigno, et al. "Neural autoregressive distribution estimation." JMLR (2016).

**Questions:**

1. (Related to weakness 1) Why should one use the proposed binary classification approach for OOD detection instead of simply approximating $p_{tr}(x)$ using, e.g., KDE or NADE?
2. Similarly, I seems feasible to use the minimum distance between a sample $x$ and samples in the in-distribution dataset $D$—often referred to as "novelty" [2] and easy to compute—as an OOD score. Have you considered this approach? If so, is there a specific reason why the proposed method (the learned binary classifier) might be more effective for OOD detection?
3. Could you explain why the proposed algorithm is called a "meta-algorithm"?
4. Recent works have attempted to inject structural biases into surrogate models to improve OOD generalization in offline MBO settings [3, 4]. It would be interesting to explore how these structurally-biased surrogate models could synergize with the proposed OOD detection method, potentially opening up future research directions.

[2] Kim, Minsu, et al. "Bootstrapped training of score-conditioned generator for offline design of biological sequences." NeurIPS (2024).
[3] Grudzien, Kuba, et al. "Functional Graphical Models: Structure Enables Offline Data-Driven Optimization." AISTATS (2024).
[4] Grudzien, Kuba, et al. "Cliqueformer: Model-Based Optimization with Structured Transformers." arXiv:2410.13106 (2024).

---

> ### Author Response · Authors · 2024-11-22
>
> We thank the reviewer for their careful review. Our response to specific comments are below:
>
> > If so, there are several ways to achieve this, e.g., kernel density estimation (KDE) or neural autoregressive density estimation (NADE; [1])
>
> Direct density estimation in high dimensional spaces such as those of biological sequences is notoriously difficult when one wants to use the numerical density values rather than sample from the learned density. Consider for example [1], where the authors find that OOD examples are often assigned higher model densities than in-distribution examples. To demonstrate these challenges, we trained a NADE model on the AAV training data and applied the log density estimates to the design selection tasks whose results are shown in Figure 3. We find that in this setting, NADE performs slightly better than the deep ensemble uncertainties, but worse than the OOD classifier (i.e. the Pearson correlation between packaging measurements and the NADE scores is -0.38 compared to -0.54 and -0.18 for the OOD classifier and deep ensemble uncertainty, respectively, and the minimum mean top 100 regret for the NADE scores is 0.65 compared to 0.18 and 1.02 for the OOD classifier and deep ensemble uncertainty, respectively)
>
> In response to another reviewer’s questions, we have also tested a baseline where we fit an Isolation Forest model [2] to ESM2 embeddings of the AAV sequences. The Isolation Forest is an established anomaly detection method that is based on the intuition that anomalous examples require fewer partitions to isolate from the rest of the data than regular examples. We find that this method works better than both the NADE scores and deep ensemble uncertainties, but worse than the OOD classifier (i.e. the Pearson correlation between packaging measurements and the Isolation Forest scores is -0.5 and the minimum mean top 100 regret for the NADE scores is 0.42). We will discuss both of these results in the main text and add the complete results to the Appendix.
>
> [1] Nalisnick, E., Matsukawa, A., Teh, Y. W., Gorur, D. & Lakshminarayanan, B. Do Deep Generative Models Know What They Don’t Know? arXiv (2018) doi:10.48550/arxiv.1810.09136.
> [2] Liu, F. T., Ting, K. M. & Zhou, Z.-H. Isolation Forest. 2008 Eighth IEEE Int. Conf. Data Min. 413–422 (2008) doi:10.1109/icdm.2008.17.
>
> > I'm uncertain how the proposed method "can significantly improve design quality" (line 27). Choosing the right threshold seems critical to effectively balance the exploitation of the surrogate model and the OOD robustness.
>
> The results in Figure 3c demonstrate that selecting sequences based on OOD scores leads to lower regret than alternative strategies, which leads to the conclusion that the method can improve the quality of designs. Indeed, the choice of threshold affects this result; we suggest in the main text (line 228) that stratifying across thresholds is a reasonable strategy to resolve this problem. Notably, a threshold percentile around 10 performs well in both the results shown in Figure 3c and the Protein Structure Prediction experiment shown in Appendix C.
>
> > The proposed algorithm relies on the MBO algorithm to generate the OOD dataset for classifier training. However, this paper only validates it using a single MBO algorithm, AdaLead, which raises concerns about the method’s versatility and generalizability.
>
> This is a consequence of performing real world biological experiments on our designed sequences. These experiments limit the number of sequences that can be tested and thus we chose to use one design algorithm to enable us to include sequences at each optimization iteration of AdaLead. To us, this is a reasonable tradeoff in order to perform real-world experiments.
>
> > A minor point, but the writing could be improved for better readability. For instance, it might be helpful to create a separate 'Preliminaries' section for the content in Sections 2.1 (offline MBO) and 2.2 (distribution shift), allowing the 'Method' section to focus solely on the main contribution
>
> We thank the reviewer for these suggestions and will incorporate them into our updated manuscript. The change suggested by the reviewer will work with the changes we have committed to in response to other reviewers to strengthen the clarity of our contribution.

---

> > ### Author Response · Authors · 2024-11-22
> >
> > > Similarly, I seems feasible to use the minimum distance between a sample x and samples in the in-distribution dataset D—often referred to as "novelty" [2] and easy to compute—as an OOD score. Have you considered this approach?
> >
> > While these novelty score can be reasonable for small datasets, their calculation requires N*M edit distance calculations, where N is the size of the training set and M is the number of designed sequences. In the AAV case, $N\approx 50,000$ and $M\approx 5,000$ and therefore calculation of novelty scores requires over 200 million edit distance calculations! Due to this unfavorable scaling, it is unfeasible to use these novelty scores as a baseline method.
> >
> > > Could you explain why the proposed algorithm is called a "meta-algorithm"?
> >
> > With this terminology, we meant to refer to the fact that the method we test is not a design algorithm itself, but can be applied as a “plug in” to any other design method. We recognize that this may be confusing when compared to meta-learning, which would be more analogous to a method that chooses or creates its own design algorithm. We will thus remove this terminology in the updated manuscript.
> >
> > > Recent works have attempted to inject structural biases into surrogate models to improve OOD generalization in offline MBO settings [3, 4]. It would be interesting to explore how these structurally-biased surrogate models could synergize with the proposed OOD detection method, potentially opening up future research directions.
> >
> > We agree that these are exciting avenues for future research, which we will mention in the Discussion section of the updated manuscript. Both of these methods require modifications to the design algorithm itself and therefore we can not test them for the current paper without re-running the in-vitro experiments.

---

> > > ### Comment · Reviewer_okPN · 2024-11-25
> > >
> > > I apologize for the late response and appreciate your effort on the rebuttal. Overall, I'm pleased with the authors' responses, but I still have some concerns.
> > >
> > > > This is a consequence of performing real world biological experiments on our designed sequences. These experiments limit the number of sequences that can be tested and thus we chose to use one design algorithm to enable us to include sequences at each optimization iteration of AdaLead. To us, this is a reasonable tradeoff in order to perform real-world experiments.
> > >
> > > I believe you could have conducted tests in the "2D Toy Model (4.1)" and the "Simulated Protein Structure Design (4.2)".
> > >
> > > > we suggest in the main text (line 228) that stratifying across thresholds is a reasonable strategy to resolve this problem.
> > >
> > > I'm not convinced that this suggestion fully addresses the issue of setting the threshold (since the suggestions still need humans to set the threshold).
> > >
> > > > Notably, a threshold percentile around 10 performs well in both the results shown in Figure 3c and the Protein Structure Prediction experiment shown in Appendix C.
> > >
> > > I believe the threshold should depend not only on the task but also on the dataset. For example, when we have a large amount of data and an accurate surrogate model, we can be more aggressive; conversely, with a small dataset and an inaccurate model, we should be more conservative.
> > >
> > > > W5
> > > 1. ***Although this is still a minor issue***, I think placing the subsections "2.1. Offline Model-Based Optimization" and "2.2. Distribution Shift in Design" under the "2. Method" section could be misleading.  Since these sections introduce previous studies and outline the problems, it might be better to position them outside of the "Method" section, where your contribution should be emphasized.
> > >
> > > 2. The verbosity has not been improved.

---

### Official Review · Reviewer_6qeD · 2024-11-04

**Soundness:** 2
**Presentation:** 2
**Contribution:** 1
**Rating:** 3
**Confidence:** 4

**Summary:**

This work proposes an out-of-distribution (OOD) classifier to detect distribution shifts, guiding design selection to avoid adversarial results. The authors suggest multiple ways to guide or filter sequence generation based on the predictions of the OOD classifier. The proposed method is tested on three different tasks, including AAV sequence design, using two different search methods, AdaLead and beam search. The experimental results show that the proposed OOD classifier achieves lower regret scores compared to deep ensemble-based OOD detection.

**Strengths:**

- I agree with the motivation that trained models can be unreliable, and handling distribution shifts is crucial in model-based optimization (MBO).
- The proposed method is straightforward and effective compared to deep ensembles
- Simplicity and ease of implementation

**Weaknesses:**

- The novelty of this approach is limited: The idea of OOD classifier is not new, and the way of using the predicted OOD score is not particularly novel neither. OOD score is simply used to filtering out the sample with threshold or range.
- The claim that "the complexity of using MBO algorithms correctly... limits the adoption among practitioners. Selecting a trust region for any search algorithm can be an art rather than a science and risks wasting experimental resources" may be overstated. Several studies have focused on discovering new sequence designs while maintaining close distances to known designs (wild types). For example, proximal exploration (PEX) [1] has made significant progress by effectively balancing the enforcement of in-distribution constraints and exploration in a practical and scientific way. Though they assume multiple query rounds in their original setting, PEX gives competitive results with a single round, which is the same as MBO. Including further discussion for other MBO approaches that consider distribution shift, such as RoMA [2] and BDI [3].

#### Minor comments
- Line 65: expansive → expensive
- Line 71: In this work, propose (there is no subject)
- For me, meta-heuristic sounds improper in this context
- In Appendix E.  Fig Xa, Fig Xb, Fig Xc

#### References
[1] Ren, Zhizhou, et al. "Proximal exploration for model-guided protein sequence design." International Conference on Machine Learning. PMLR, 2022.

[2] Yu, Sihyun, et al. "Roma: Robust model adaptation for offline model-based optimization." Advances in Neural Information Processing Systems 34 (2021): 4619-4631.

[3] Chen, Can, et al. "Bidirectional learning for offline infinite-width model-based optimization." Advances in Neural Information Processing Systems 35 (2022): 29454-29467.

**Questions:**

- From my understanding, the optimization process seems to be conducted iteratively. Does it means multiple query rounds like the setting in AdaLead? If not, please clearly state the difference with AdaLead setting. If yes, the motivation and approach might be improper (even though I agree with the claim that we should carefully handle the unreliable surrogate model for the adversarial samples, as mentioned above), as a key assumption in MBO is that we cannot make additional queries to the black-box function. Allowing additional queries can lead to significant differences in the methodologies used. For instance, we need to explore the unreliable region for the subsequent iterations rather than filtering out these samples.
- Regarding the AAV task, is the surrogate model the same as the one used in AdaLead? If not, I am concerned that the comparison between the OOD classifier and deep ensembles might not be entirely fair. I have checked Appendix A.2, but I am unsure whether the model capacity is sufficient to learn the fitness function of the AAV tasks and, consequently, whether deep ensemble-based OOD detection would be effective with an insufficiently trained surrogate model.
- How many models are used for the deep ensemble?
- What is the meaning of "50 bootstrap data samples" in line 472?
- Additionally, I am curious whether the proposed OOD classifier could benefit search methods that already enforce in-distribution samples, such as PEX.

---

> ### Author Response · Authors · 2024-11-22
>
> We thank the reviewer for their feedback and respond to specific comments below:
>
> > The novelty of this approach is limited: The idea of OOD classifier is not new, and the way of using the predicted OOD score is not particularly novel neither. OOD score is simply used to filtering out the sample with threshold or range.
>
> We do not intend to claim that the use of a binary classifier to detect covariate shift is novel, and have cited a number of papers that use such methods. Our contributions are two fold: first, we demonstrate the extent to which feedback covariate shift (FCS) is present in a real-world sequence design problem and how it can impact the resulting design. Second, we demonstrate that an application of the OOD classifier to distinguish between the training and designed distribution can mitigate the effects of FCS.
>
> We are committed to clarifying our contributions by modifying the writing of the paper in a number of ways. First, we will modify the introduction to emphasize the two contributions discussed above. Second, we will further emphasize the previous work that has used binary classifiers for covariate shift detection. Finally, we will move the derivation of the “density ratio trick” in Section 2.3 to the SI, as this section may be giving an unintended impression of novelty.
>
> > "the complexity of using MBO algorithms correctly... limits the adoption among practitioners. Selecting a trust region for any search algorithm can be an art rather than a science and risks wasting experimental resources" may be overstated. For example, proximal exploration (PEX) [1]...
>
> PEX is indeed a powerful method for designing biological sequences when one starts only with the wild type (WT) sequence and is able to make a number of experimental queries (i.e. in an online setting). In contrast, our paper is focused on the more common offline setting where one has an existing dataset of sequence-function measurements for sequences with a variety of edit distances to WT and is designing sequences for only one follow-up experiment. PEX primarily uses a constraint on the edit distance to WT to limit the distribution shift of the designed sequences. In the offline setting, this edit distance constraint will not take into account the distribution of the existing data and thus will be overly restrictive in the case where there are high edit distance sequences in the existing data (as in our case; see Figure 4 in Appendix A). PEX is thus not an applicable baseline to our problem. Nonetheless, we agree with the reviewer that the language in this sentence is too strong and we will soften it to recognize contributions to this field.
>
> > Including further discussion for other MBO approaches that consider distribution shift, such as RoMA [2] and BDI [3].
>
> We thank the reviewer for providing these citations and we will add them to our discussion in the Related Work section. Notably, both of these methods require modifications to the surrogate models and design algorithm. In contrast, we discuss a method that can be used as a “drop-in” with any design method, given only training and designed sequences. It can thus be more practical in many design scenarios.
>
> > From my understanding, the optimization process seems to be conducted iteratively. Does it means multiple query rounds like the setting in AdaLead?
>
> We assume the reviewer is asking whether the optimization procedure is online or offline (i.e. that “iteratively” refers to multiple rounds of experimental query). Our paper is focused on the offline setting, where the ground truth experiment cannot be queried during the design procedure. This is also the case for AdaLead, which is offline genetic that queries the **surrogate model** during iterative rounds of mutation and selection. We believe the misunderstanding in how AdaLead is used may arise from the PEX paper, where (we believe) the authors query the ground truth experiment during the genetic algorithm iterations, rather than the surrogate model. It is perfectly valid to use AdaLead in this way, but is not the way that it was intended to be used.
>
> > Regarding the AAV task, is the surrogate model the same as the one used in AdaLead?
>
> Yes, the surrogate model used in AdaLead is the mean of the ensemble used to calculate ensemble uncertainties. We will clarify this in the Appendix.
>
> > How many models are used for the deep ensemble?
>
> We used 10 models with independent initializations for the ensemble. We ensure that this detail is added the main text.
>
> > What is the meaning of "50 bootstrap data samples" in line 472?
>
> In this case, the designed sequences are fixed since they were experimentally tested prior to the application of our selection method. In order to ensure that our regret calculations were not anomalous to the specific set of designed sequences, we resampled the designed sequences with replacement 50 times and calculated the regret curves for each resampling. We will clarify this procedure in the paper.

---

> > ### Comment · Reviewer_6qeD · 2024-11-25
> >
> > Thank you for your effort, and I apologize for the delayed response. I appreciate the detailed explanations you've provided. However, I still have some concerns I'd like to discuss:
> >
> > - From what I understand, the OOD classifier is applied to distinguish between the training and designed distributions to enhance MBO methods. It filters out OOD samples, resulting in lower regret scores compared to the Ensemble method. My concern is whether this approach ultimately benefits the generation of more desirable samples (e.g., higher f) or more diverse and novel batched samples. Since these metrics are commonly reported in various studies, could you provide these results, at least for the simulated protein design?
> >
> > - The manuscript can be revised during the discussion period. Any updates or clarifications you can provide would help me reconsider my review.

---

### Official Review · Reviewer_eBHJ · 2024-11-09

**Soundness:** 2
**Presentation:** 2
**Contribution:** 1
**Rating:** 3
**Confidence:** 3

**Summary:**

This paper introduces a method to detect and correct for feedback covariate shift in experimental design (such as protein sequence design), where training data distribution differs from the distribution of the new data candidates generated throughout the process. The method is based on softmax regression for binary classification of the domain, which is employed to discriminate between the training data and the data generated within the feedback loop. The logit score (here OOD score) represents the intensity of the covariate shift. The authors empirically validate their approach on three use-cases: synthetic function, biological simulation, and real-world application of protein sequence design. The domain classifier equipped with the OOD score is able to identify and reduce the distribution shift in the design loop in a real-world application.

**Strengths:**

The paper presents an application in biology of the method for addressing feedback covariate shift. It is indeed of crucial importance to be able to correct for this shift across many fields with the strong presence of automated experimental design, biology being only of them. The method is very simple to integrate into any existing experimental pipeline, and seems to be performing well in practice.

**Weaknesses:**

To the best of my knowledge, I believe this way of addressing covariate shift is not novel; the novelty might lie in the aspect of applying this method to *feedback* covariate shift (see for example, https://doi.org/10.7551/mitpress/9780262170055.003.0008, or https://dl.acm.org/doi/10.5555/1577069.1755858, and references therein). Domain classification by means of logistic regression (with or without importance weighting) is one of the widely known methods, hereby just applied to the special case of feedback covariate shift in experimental design. In feedback covariate shift, it is assumed that the distribution of points generated within the feedback loop depends also on the training distribution. The paper lacks a clearer presentation of its methodological contributions, and does not fully allow one to appreciate the usefulness of the method in practice. Comparison with other baselines (e.g., other methods for unsupervised domain adaptation) would further highlight the strengths and weaknesses of the method. The running time of the method was not investigated, i.e., how much of the optimization budget needs to be dedicated to detecting and correcting for shifts in a real-world application. This could be done by comparing the computational overhead of the proposed method to the baseline optimization approach in real-world scenarios. The figures throughout the paper are not necessarily self-explanatory. Furthermore, a more rigorous technical and mathematical notation is missing.

**Questions:**

Could you please clarify how your approach differs from or improves upon existing methods for addressing covariate shift, particularly in the context of feedback covariate shift in experimental design? It would help to highlight your exact contributions more clearly and position with greater care your approach within related work.

---

> ### Author Response · Authors · 2024-11-22
>
> We thank the reviewer for the thoughtful and constructive feedback. We address specific comments below:
>
> > To the best of my knowledge, I believe this way of addressing covariate shift is not novel; the novelty might lie in the aspect of applying this method to feedback covariate shift
>
> Indeed, we do not intend to claim that using logistic regression to address covariate shift is novel and we cite a number of papers that use this technique. Our contributions are two fold: first, we demonstrate the extent to which feedback covariate shift (FCS) is present in a real-world sequence design problem and how it can impact the resulting design. Second, we show that we can mitigate the effects of FCS by training the logistic regression model to distinguish between the training and designed distributions and then using the log probability scores of the model to select sequences that are less likely to have been affected by FCS. Our paper is thus impactful for practitioners of sequences, as it highlights the dangers of FCS in this scenario and provides a straightforward technique for mitigating its worst effects.
>
> We are committed to clarifying our contributions by modifying the writing of the paper in a number of ways. First, we will modify the introduction to emphasize the two contributions discussed above. Second, we will further emphasize the previous work that has used logistic regression for covariate shift detection and add additional citations, including those given by the reviewer. Finally, we will move the derivation of the “density ratio trick” in Section 2.3 to the SI, as this section may be giving an unintended impression of novelty.
>
> > Comparison with other baselines (e.g., other methods for unsupervised domain adaptation) would further highlight the strengths and weaknesses of the method.
>
> We use the logistic regression method to remove designed sequences with unreliable predicted fitness due to FCS. We are therefore most concerned with the ability to detect whether individual points are distant from the training distribution and thus likely to be affected by FCS. This is a different problem then is typically approached with unsupervised domain adaption approaches, where one is typically concerned with detecting and adapting to distributional shifts between the training and test distributions. The most salient baselines to compare to the logistic regression method are instead methods for anomaly detection. One such method is the Isolation Forest [1],  an established anomaly detection method that is based on the intuition that anomalous examples require fewer partitions to isolate from the rest of the data than regular examples. We implemented a version of the Isolation Forest for anomaly detection in sequences where the sequences are first embedded into a continuous space using the ESM2 protein language model with mean pool embedding and then an Isolation Forest is trained on the resulting embeddings. We applied this method to the AAV design problem and found that it performed better than ensemble uncertainties, but slightly worse than the OOD classifier (i.e. the Pearson correlation between packaging measurements and the Isolation Forest scores is -0.50 compared to -0.54 and -0.18 for the OOD classifier and deep ensemble uncertainty, respectively, and the minimum top 100 regret for the Isolation Forest scores is 0.42 compared to 0.18 and 1.02 for the OOD classifier and deep ensemble uncertainty, respectively). In response to another reviewer’s questions, we have also tested a baseline where we fit a Neural Autoregressive Density Estimation (NADE) [2] model and used the resulting density estimates as scores to select sequences. We find that this method again works better than the deep ensemble uncertainties, but worse than the Isolation Forest scores and OOD classifier scores  (i.e. the Pearson correlation between packaging measurements and the NADE scores is -0.38 and the minimum mean top 100 regret for the NADE scores is 0.65). We will discuss both of these results in the main text and add the complete results to the Appendix.
>
> [1] Liu, F. T., Ting, K. M. & Zhou, Z.-H. Isolation Forest. 2008 Eighth IEEE Int. Conf. Data Min. 413–422 (2008) doi:10.1109/icdm.2008.17.
> [2] Uria, Benigno, et al. "Neural autoregressive distribution estimation." JMLR (2016).

---

> > ### Author Response · Authors · 2024-11-22
> >
> > > The running time of the method was not investigated, i.e., how much of the optimization budget needs to be dedicated to detecting and correcting for shifts in a real-world application
> >
> > Our paper is written in the context of offline model-based optimization (MBO). These types of problems are “offline” because the temporal and dollar cost of collecting experimental data is much higher than the cost of computational design. In these cases, the specific computational complexity of design methods is not particularly important. For example, in the AAV case, the in-vitro experiments can take months to perform whereas the design can be done in a matter of minutes or hours with basic computational resources. For this reason, we do not perform a rigorous run time analysis of the method. However, we will highlight that the method only requires training a dense MLP on one-hot encoded sequences which can be done in a matter of minutes on a single T4 GPU.

---

### Meta-Review · Area_Chair_X7yx · 2024-12-21

**Metareview:**

The paper introduces a binary classification-based meta-heuristic to address distribution shift in model-based optimization (MBO), with applications to biological sequence design. While the idea is relevant and straightforward to implement, reviewers highlighted key weaknesses, including limited novelty, insufficient experimental validation across diverse MBO frameworks, and inadequate exploration of threshold selection and generalizability. The paper also lacked a clear demonstration of how the method improves design quality beyond OOD detection and was limited to testing with a single MBO algorithm, raising concerns about versatility. Although the authors provided clarifications and additional comparisons to alternative methods, these responses did not fully address the primary concerns. Consequently, the paper does not meet the standards for acceptance due to its limited scope, lack of rigorous novelty, and insufficient validation.

**Additional Comments On Reviewer Discussion:**

During the rebuttal period, reviewers raised concerns about the paper's novelty, limited experimental scope, and unclear contributions.

Reviewer *eBHJ* questioned the novelty of using a binary classifier for OOD detection and requested comparisons with baseline methods for unsupervised domain adaptation. The authors clarified their contributions and added comparisons to Isolation Forest and Neural Autoregressive Density Estimation, but these additions highlighted only marginal improvements.

Reviewer *6qeD* critiqued the lack of task diversity in experiments and the reliance on a single MBO algorithm, while the authors argued this was due to constraints from real-world biological experiments.

Reviewer *okPN* questioned the practicality of threshold selection and the lack of generalizability testing, to which the authors proposed stratified thresholding and noted the need for future research. Reviewer 9umf acknowledged the method’s straightforward implementation but requested guidance on leveraging OOD scores beyond binary filtering, which was only partially addressed.

Overall, while the rebuttal clarified some points and improved the manuscript, the core concerns about limited validation, novelty, and generalizability remained unresolved, leading to the decision to reject.

---

### Decision · Program_Chairs · 2025-01-22

Reject